# Overexpression and Activation of αvβ3 Integrin Differentially Affects TGFβ2 Signaling in Human Trabecular Meshwork Cells

**DOI:** 10.3390/cells10081923

**Published:** 2021-07-29

**Authors:** Mark S. Filla, Kristy K. Meyer, Jennifer A. Faralli, Donna M. Peters

**Affiliations:** 1Pathology & Laboratory Medicine, University of Wisconsin, Madison, WI 53705, USA; msfilla@wisc.edu (M.S.F.); kristymeyer@wisc.edu (K.K.M.); peters10@wisc.edu (J.A.F.); 2Ophthalmology & Visual Sciences, University of Wisconsin, Madison, WI 53705, USA

**Keywords:** trabecular meshwork, glaucoma, integrins

## Abstract

Studies from our laboratory have suggested that activation of αvβ3 integrin-mediated signaling could contribute to the fibrotic-like changes observed in primary open angle glaucoma (POAG) and glucocorticoid-induced glaucoma. To determine how αvβ3 integrin signaling could be involved in this process, RNA-Seq analysis was used to analyze the transcriptomes of immortalized trabecular meshwork (TM) cell lines overexpressing either a control vector or a wild type (WT) or a constitutively active (CA) αvβ3 integrin. Compared to control cells, hierarchical clustering, PANTHER pathway and protein-protein interaction (PPI) analysis of cells overexpressing WT-αvβ3 integrin or CA-αvβ3 integrin resulted in a significant differential expression of genes encoding for transcription factors, adhesion and cytoskeleton proteins, extracellular matrix (ECM) proteins, cytokines and GTPases. Cells overexpressing a CA-αvβ3 integrin also demonstrated an enrichment for genes encoding proteins found in TGFβ2, Wnt and cadherin signaling pathways all of which have been implicated in POAG pathogenesis. These changes were not observed in cells overexpressing WT-αvβ3 integrin. Our results suggest that activation of αvβ3 integrin signaling in TM cells could have significant impacts on TM function and POAG pathogenesis.

## 1. Introduction

Glaucoma is a chronic optic neuropathy that results in irreversible blindness [1]. Although the exact etiology of glaucoma remains unknown, a number of studies have now determined that elevated levels of transforming growth factor β2 (TGFβ2) and connective tissue growth factor (CTGF) and the use of glucocorticoids can lead to the development of glaucoma [2,3]. Clinically, glaucoma induced by TGFβ2, CTGF and glucocorticoids (GC) is very similar. All cause an increase in extracellular matrix (ECM) deposition and cytoskeleton changes associated with contractility that lead to a restriction in aqueous humor outflow and together contribute to the elevation of intraocular pressure (IOP) associated with glaucoma.

Recent studies have shown that the activation of αvβ3 integrin signaling plays a role in regulating these changes in ECM deposition and contractility. Activation of the integrin enhances the deposition of EDA+ fibronectin into the ECM [4] which in turn can trigger profibrotic changes in the ECM and elevation in IOP [5,6]. The αvβ3 integrin also promotes the formation of crosslinked actin networks (CLANs) that may cause the contractility changes in the TM thought to be associated with a stiffer fibrotic-like tissue [7,8,9,10]. Activation of αvβ3 integrin also increases IOP in mice and porcine organ cultured anterior segments. In contrast, knockdown of the αvβ3 integrin gene and its expression in the TM and ciliary muscle (CM) using a tamoxifen inducible Cre +/-β3 integrin flox/flox mouse model decreases IOP in mice [11]. The activity of the αvβ3 integrin in human trabecular meshwork (HTM) cells can be increased by the glucocorticoid dexamethasone via the calcineurin/NFATc1 pathway [12]. This pathway has been associated with glaucoma since it regulates the expression of myocilin, a protein involved in an early onset form of POAG and GC-induced ocular hypertension [13]. Together these results suggest that αvβ3 integrin signaling can contribute to the development of glaucoma.

αvβ3 integrin is a member of the integrin family of transmembrane receptors that mediate adhesion to the ECM. In addition to their mechanical roles in mediating adhesion, integrins transmit chemical signals into the cell that provided information on a cell’s local environment [14]. These signals determine a cell’s ability to respond to other signals transmitted by growth factor- or G-protein-coupled receptors. Some of the major pathways co-regulated by integrins include PI3K/Akt, MAPK/Erk, Jnk, and Rho GTPase [15]. In addition, integrins can mediate gene expression independent of their adhesion role [16,17]. Thus, activation of integrins has the potential to play several critical roles in regulating signaling pathways proposed to be altered in glaucoma. 

To understand how αvβ3 integrin could affect the signaling pathways associated with glaucoma in the trabecular meshwork, RNA-Seq analysis was used to study changes in the transcriptome of TM cells when αvβ3 integrin was activated. The study used the immortalized TM-1 cell line in which either a wild type (WT) or a constitutively active (CA) form of the αvβ3 integrin was overexpressed [18]. TM-1 cells transduced with an empty vector were used as a control. The differential expression of genes between these cell lines was analyzed using various bioannotations, network and pathway analysis databases to determine specific signaling pathways and categories of genes that may be altered when αvβ3 integrin is active. 

## 2. Materials and Methods

### 2.1. Cell Culture

Human SV40-immortalized TM-1 cells where generated from primary human TM (HTM) cells [18] that were derived from a 30-year-old donor eye as previously described [19,20]. To the best of our knowledge, the donor eye issue was free of any ocular diseases. The isolated parent HTM cells showed morphological features expected of human trabecular meshwork cells [19,20] and expressed elevated levels of myocilin in response to glucocorticoids [21]. The TM-1 cells generated from these primary HTM cells appeared similar to HTM cells and meet the criteria outlined in Keller, et al. [22]. They upregulated expression of both fibronectin and myocilin mRNA in response to glucocorticoids [23] and were capable of forming CLANs [8]. TM-1 cells overexpressing a control empty vector (EV), a wild type αvβ3 integrin (WTβ3) or a CA-αvβ3 integrin (CAβ3) that contains a T562N mutation in the β3 integrin subunit [24] were subsequently generated as described by Gagen, et al. [25]. All the cell lines were cultured in low glucose DMEM (MilliporeSigma, St. Louis, MO, USA), containing 10% fetal bovine serum (FBS, Atlanta Biologicals/R&D Systems, Flowery Branch, GA, USA), 2 mM L-glutamine (MilliporeSigma), 0.2% Primocin (Thermo Fisher Scientific, Waltham, MA) and 0.05% gentamicin (Corning, Glendale, AZ, USA). Cells were kept under selective pressure using 2 µg/mL puromycin.

### 2.2. RNA Isolation

EV, WTβ3 and CAβ3 cells were plated into P-60 dishes at 3.5 × 10^6^ cells/dish. Four dishes were plated for each cell line. Cells were plated in low serum (1% FBS) containing medium and allowed to attach overnight. Twenty-four hrs post-plating, cells were lifted with trypsin, washed once with phosphate-buffered saline (PBS) and then pelleted by centrifugation. Pellets were frozen and then stored at −20 °C until processed for RNA isolation. RNA was isolated from each pellet using the RNeasy Micro kit with QIAshredder (Qiagen, Germantown, MD, USA) according to the manufacturer’s instructions.

### 2.3. RNA-Seq Analysis

RNA libraries were prepared using 1µg of RNA/sample with the TruSeq RNA sample preparation kit (Illumina, San Diego, CA, USA) and sequenced on four lanes of an Illumina HiSeq 2500 System. The Illumina RNA library preparation and sequencing was done by the University of Wisconsin Biotechnology Center Gene Expression Center. Sequencing reads were checked for quality using the fastqc program (Babraham Bioinformatics, Babraham Institute, Cambridge, UK). The trimming software skewer [26] was used to preprocess raw fastq files. After trimming, the UW-Madison Bioinformatics Resource Center pipeline was used to compute the QC statistics. The QC statistics pipeline included the following: (1) combined per cycle base quality, (2) per cycle base frequencies, (3) per cycle average base quality, (4) relative 3k-mer diversity, (5) Phred quality distribution, (6) mean quality distribution, (7) read length distribution and (8) read occurrence distribution. The reads were aligned against the reference genome using STAR (Spliced Transcripts Alignment to a Reference) [27]. Low-abundance genes with a read count < 1 Log CPM (counts per million) in two or more samples were excluded. Each RNA-Seq sample was normalized by the method of trimmed mean M-values [28]. Identification of the differentially expressed genes was annotated using Ensembl Genes version GRCH38.p12 [29]. Differentially expressed genes were considered significant with an adjusted *p*-value < 0.05 based on the false discovery rate (FDR) using the Benjamini-Hochberg (BH) procedure and the logarithm of fold change (FC). Three paired contrasts were performed for the differential gene expression (DGE) analysis: WTβ3 vs. EV cells (WTβ3-EV), WTβ3 vs. CAβ3 cells (WTβ3-CAβ3) and CAβ3 vs. EV cells (CAβ3-EV). The edgeR contrast data for all three cell lines that provided the basis for this publication can be accessed in the Appendix A Those genes that did not demonstrate a significant change as determined by DGE analysis have been removed from this file. However, the complete sequencing data, the processed data used to compute the log_2_ fold changes as well as the complete DGE analysis data from all three contrasts have been deposited in NCBI’s Gene Expression Omnibus and are accessible through GEO Series. Accession number: GSE180407.

### 2.4. RT-qPCR Analysis

cDNA was generated with the High-capacity cDNA reverse transcription kit (Thermo Fisher Scientific) according to the manufacturer’s instructions. qPCR was performed using an Applied Biosystems QuantStudio 6 Pro Real-Time PCR system and PowerUp SYBR green master mix (Thermo Fisher Scientific) as we described [23]. Fold changes in gene expression were determined using the ΔΔCt method. Data were normalized to the housekeeping gene succinate dehydrogenase complex subunit A (SDHA). Primer-BLAST (National Center for Biotechnology Information (NCBI), Bethesda, MD, USA) was used to design the primers which were made by IDT (Integrated DNA Technologies, Inc., Coralville, IA, USA).

### 2.5. R Analysis

R statistical computing and graphics software was used to identify unique or shared differentially expressed genes in WTβ3 and CAβ3 cells that demonstrated a ±1.5 Log_2_ FC. VennDiagram, readxl, tidyverse and dplyr R packages were used for this purpose. Generation of heat maps was performed using readxl, dplyr and gplots R packages. In some instances, the FC of a gene in one of the three contrasts was missing from the dataset because that gene demonstrated a subthreshold read count in one of the three cell lines in the RNA-Seq analysis (see above). In order to display all three contrasts for these genes within the heat maps, the missing FCs were calculated in Microsoft Excel using the pre-filtered FPKM (Fragments Per Kilobase of transcript per Million) counts from the raw RNA-Seq data using the following formula:$$\;Log_{\begin{array}{c} 2 \end{array}}FC=LOG\left(\frac{FPKM1}{FPKM2},2\right)$$

In these instances, the Log_2_ FC was manually calculated for all three contrasts so the calculations would be consistent within the heat map. These genes are identified by asterisks in the heat maps.

### 2.6. TGFβ2 ELISA Analysis 

TM-1 EV or CAβ3 cells were plated at a density of 2.0 × 10^5^ cells/4 cm^2^ in growth medium containing 10% FBS and allowed to attach overnight. Cells were incubated for 24 hr in medium containing 1% FBS. The conditioned media (CM) was collected and debris removed via centrifugation. The CM was acidified by addition of 1N HCL for 10 min in order to activate the TGFβ2. The CM was then neutralized by addition of NaOH in 0.5 M HEPES. ELISA analysis was performed using an R&D Systems Human TGF-beta 2 Quantikine ELISA Kit (R & D Systems, Minneapolis, MN), and the procedure was performed according to the manufacturer’s instructions. Samples were read on a BioTek Synergy Neo2 plate reader (Agilent, Santa Clara, CA, USA) at 450 nm. Statistical analysis of the data was performed using analysis of variance (ANOVA).

### 2.7. Bioannotations, Network and Pathway Analysis 

Networks were generated using the STRING (Search Tool for the Retrieval of Interacting Genes/Proteins) database v. 11.0b [30] to identify known and predicted protein-protein interactions. Medium (0.4) and high (0.7) confidence networks were both examined. The PANTHER (Protein Analysis Through Evolutionary Relationships) database v. 16.0 [31] was also used for pathway analysis. A statistical overrepresentation test (Fisher’s Exact test) was used to determine if any metabolic or signaling pathways were enriched in the WTβ3 or CAβ3 cell lines.

Bioannotations of the differentially expressed genes used KEGG (Kyoto Encyclopedia of Genes and Genomes) v. 96.0 [32,33,34], DAVID Bioinformatics Resources v. 6.8 [35,36] the Matrisome Project database v. 2.0 [37] and UniProt v 2020 06 [38] databases for gene classifications. Non-coding genes and transcripts were classified using the GeneCards database [39]. In some instances, the EMT-Regulome database v. 1.0 [40] was used to examine the role(s) of certain genes in regulating epithelial-mesenchymal transition (EMT).

## 3. Results

### 3.1. αvβ3 Integrin Alters the Expression Pattern of the TM Transcriptome 

Since activation of αvβ3 integrin signaling can trigger changes linked to glaucoma [41], RNA-Seq analysis was performed to determine how activation of αvβ3 integrin signaling affects the transcriptome of TM cells. Figure 1 shows unsupervised multidimensional scaling (MDS) of the genes (1A) and transcripts (1B), respectively, in cells that overexpressed either a CA-αvβ3 integrin (CAβ3 cells), a WT-αvβ3 integrin (WTβ3 cells) or a control empty vector (EV cells) that lacked αvβ3 integrin. Both MDS plots show that, relative to the EV cell line, both the WTβ3 and CAβ3 cell lines represented distinct populations of cells with specific gene transcripts being upregulated or downregulated by the expression and/or activity of αvβ3 integrin. Similarly, the WTβ3 and CAβ3 cell lines were found to be distinct relative to each other. Furthermore, these plots showed that there was little variance within the four samples from each cell line. Together these data show that the differences between the cell lines were larger than the differences within each cell line. 

Comparing the transcriptomes of the three cell lines found significant DGE between them (Table 1). The transcriptome of WTβ3 cells showed a statistically significant (*p* < 0.05) difference in 5186 genes compared to that of control EV cells. Of those altered genes, 2645 were upregulated in WTβ3 cells while 2541 were downregulated. We also found a statistically significant difference (*p* < 0.05) in 8249 genes in the transcriptome of CAβ3 cells compared to EV cells. In this case, 4094 genes were upregulated in CAβ3 cells while 4255 were downregulated. The transcriptome of WTβ3 cells compared to CAβ3 cells also differed. The WTβ3 transcriptome showed a significant difference (*p* < 0.05) in 6999 genes compared to that of CAβ3 cells (3524 upregulated; 3475 downregulated). 

### 3.2. RNA-Seq Analysis Reveals Genes Enriched or Depleted by Activation of αvβ3 Integrin Signaling 

We then identified genes that demonstrated a ± 1.5 Log_2_ FC in expression in the WTβ3 and CAβ3 cell lines compared to the control EV cells (Appendix A). As shown in Figure 2, overexpression of the WT-αvβ3 integrin caused an enrichment in 113 genes and a depletion in 73 genes. In contrast, expression of the CA-αvβ3 integrin caused an enrichment in 273 genes and a depletion in 149 genes. When comparing the gene expression changes, we saw that 65 genes were enriched in both cell lines while 34 genes were depleted in both cell lines. This suggests that activation of αvβ3 integrin signaling differentially affected the TM-1 transcriptome compared to mere overexpression of WT-αvβ3 integrin.

This differential expression was detected in both the coding and non-coding genomic regions of both cell lines. A total of 447 genes were detected from the coding regions (Appendix A) while 62 transcripts were detected from non-coding regions (Appendix A). This latter group included lncRNA, lincRNA, microRNA, snRNA, snoRNA, ribosomal RNA, novel transcripts, open-reading frames, pseudogenes and antisense genes. 

To identify biological processes that may be enriched by overexpression of either WT-αvβ3 integrin or CA-αvβ3 integrin, gene ontology enrichment analysis and gene classification was performed on the genes listed in Appendix A. As shown in Figure 3, these analyses showed that multiple biological processes were affected. The top biological process affected was transcription, followed by ECM, cell adhesion and cytokine signaling. Interestingly, these biological processes were differentially affected in CAβ3 cells compared to WTβ3 cells.

Since integrin signaling pathways are key regulators of gene expression, it was not surprising that activation of αvβ3 integrin signaling caused a significant change in the expression of transcription factors and their co-activators. Genes for transcription factors and co-activators that were preferentially altered in CAβ3 cells represented 13% (43/323) of the differentially expressed CAβ3 cell transcriptome when compared to the EV cell transcriptome. In contrast, only 6% (5/87) of the genes for transcription factors and co-activators were altered in WTβ3 cells (Figure 3). The transcriptome of genes for cytokines and cytokine-related proteins was also preferentially altered in CAβ3 cells representing 6% (20/323) of the CAβ3 transcriptome. In contrast, the transcriptome of WTβ3 cells showed only a 3% (3/87) shift in genes for this group. Both transcriptomes showed similar enrichments in genes for cell adhesion receptors, especially members of the cadherin family, and represented 6% (20/323) of the transcriptome in CAβ3 cells and 7% (6/87) of the transcriptome in WTβ3 cells. Likewise, the transcriptome of ECM-associated proteins appeared to be similarly affected in both cell types and represented 7% (21/323) of the transcriptome in CAβ3 cells and 8% (7/87) of the transcriptome in WTβ3 cells. Given that integrin signaling is known to regulate the organization of the cytoskeleton, it was somewhat surprising that only 3% of the transcriptome affected by the activation of αvβ3 integrin signaling involved cytoskeleton-associated proteins and GTPases. Not surprisingly, genes in the transcriptomes of both cell lines that were commonly shared showed an enrichment in all these groups as well (see Appendix A). 

### 3.3. Expression of Transcription Factors Is Differentially Regulated in CAβ3 and WTβ3cells

Hierarchical clustering of genes encoding transcription factors and other transcription-associated proteins in the ±1.5 Log_2_ FC cutoff (Figure 4) shows the differential expression of these genes in cells overexpressing WT αvβ3 integrin compared to cells overexpressing CA αvβ3 integrin. A major difference appears to be in the expression of ZNF (zinc finger) transcription factors. Surprisingly, a total of 11 ZNF transcription factor genes demonstrated significantly altered expression in either CAβ3 or WTβ3 cells when contrasted to control EV cells (Appendix A). In CAβ3 cells, 9/11 were uniquely upregulated and 0/11 uniquely downregulated while in WTβ3 cells there was 1/11 uniquely upregulated and 1/11 uniquely downregulated. Although the CAβ3 transcriptome showed an enrichment in genes encoding members of the ZNF family of transcription factors, the significance of this is unknown since their functions are still largely unknown.

The gene clusters found within regions L-P of the Figure 4 heat map were of particular interest given their marked differential expression when gene changes in CAβ3 cells were compared to gene changes in WTβ3 cells (WTβ3-CAβ3 column). Genes within clusters L and M (*SOX5, ZIC5, MYCL, LHX2, EBF1, MYOCD, STAT5A, LHX1*) were all significantly downregulated in WTβ3 cells but were significantly upregulated in CAβ3 cells (Appendix A). In contrast, genes within clusters N-P (*GSPT2*, *RFX8*, *SPOCD1*, *RELB, SP140, FOSL1, EYA2, MAFF, BNC1, KLF4, KLF2, HCLS1)* were downregulated to a greater degree in CAβ3 cells than in WTβ3 cells (see WTβ3-CAβ3 column and Appendix A). 

Several genes encoding transcription factors in those clusters are of particular interest. For instance, *MYOCD* (CAβ3 FC = 2.94), which is upregulated in CAβ3 cells, encodes a transcriptional coactivator of serum response factor (SRF) [42] that regulates smooth muscle contraction and is involved in fibrosis [43]. *SOX5* [44] (CAβ3 FC = 3.49)*, STAT5A* [45] (CAβ3 FC = 3.32)*,* and *FOSL1* [46] (CAβ3 FC = −1.98) are also associated with fibrosis as is *KLF4* (FC = −2.58) which encodes an inhibitor of TGFβ signaling [47]. Additionally, the KLF4 transcription factor is responsive to shear stress, downregulates SPARC expression in the TM and plays a role in TM stem cell development and proliferation [48,49,50]. *KLF2* (CAβ3 FC = −1.54), also encodes an inhibitor of TGFβ signaling [51] that can function as a flow responsive transcription factor that induces endothelial nitric oxide synthase (eNOS) expression [52]. 

Other genes of interest are *PAX6*, *LMX1B* and *DDIT3*. *PAX**6* (FC = 2.78) is significantly upregulated in CAβ3 vs. WTβ3 cells like the genes in clusters L and M. It has been shown to play a role in anterior segment dysgenesis (ASD) [53] and is linked to glaucoma (Table 2). *LMX1B* (CAβ3 FC = 1.65) encodes a transcription factor that is associated with nail patella and has been shown to be associated with the development of POAG (Table 2). *DDIT3* encodes a pro-apoptotic transcription factor, also known as CHOP, that participates in regulating IOP (Table 3) and was down regulated in both WTβ3(FC = −2.90) and CAβ3 (FC = −2.25) cells.

In contrast to the numerous genes whose expression was significantly altered by constitutive activation of αvβ3 integrin in CAβ3 cells, the expression of very few transcription factors was significantly altered by overexpression of WT αvβ3 integrin in WTβ3 cells. Of those that met the FC cutoff, only *ZNF626* (FC = 1.93) was upregulated while *RUNX1T1* (FC = −2.51), *NUPR1* (FC = −2.47) and *EGR2* (FC = −1.52) were downregulated. This suggests that overexpression of WT αvβ3 integrin by itself did not have a major impact on the transcriptional activity of TM cells.

### 3.4. Hierarchical Clustering Reveals Differences in the Expression of ECM Protein Genes 

Another group of proteins that appeared to be differentially affected by expression of αvβ3 integrin and its activation were those involved in the organization of the ECM (Figure 5). Interestingly, there were only six core ECM proteins that showed differential expression in either WTβ3 or CAβ3 cells compared to EV cells that met the ±1.5 Log_2_ FC cutoff. Some of these genes are also linked to glaucoma (Table 2). Of those, 4/6 were significantly altered only in WTβ3 cells while 2/6 were significantly altered only in CAβ3 cells. Hierarchical clustering showed that overexpression of WT αvβ3 integrins caused an upregulation in *COL8A2* (FC = 2.71), *EMILIN1*(FC = 1.51) and *POSTN* (FC = 3.88). Both *COL8A2* and *POSTN* have been reported to be associated with glaucoma (Table 2). *POSTN* plays a role in ASD and disruption in its expression leads to a disorganization of collagen fibrils in the TM. Additionally, its expression has also been associated with several fibrotic diseases [73]. *COL8A2* mutations have been found in some glaucoma patients [74], and its expression is elevated in HTM cells in response to dexamethasone suggesting a role in GC-induced glaucoma as well as in the development of the anterior segment [75]. In contrast, *COL11A1* (FC = −1.96), which is associated with both POAG and primary angle closure glaucoma (PACG) (Table 2) was downregulated in WTβ3cells. Regarding core ECM genes that were significantly altered only in CAβ3 cells, neither *VWA5A* (FC = 2.67) nor *COL13A1* (FC = −1.64) have been found to be associated with glaucoma, although *COL13A1* was differentially regulated in response to the steroid triamcinolone in cultured HTM cells [61].

In addition to these core ECM protein genes, we noted the differential expression of four matricellular genes. *SPOCK1* (FC = 1.93), *TLL1* FC = 4.19) and *SULF1* (FC = 1.76) are all uniquely upregulated in CAβ3 cells. *SPOCK1* encodes a member of the SPARC family and is associated with glaucoma (Table 2). *TLL1* and *SULF1* have no reported associations with glaucoma or IOP regulation, however, both genes encode proteins associated with fibrosis and an epithelial to mesenchymal transition (EMT) [76] which shows similar phenotypic changes to that observed in POAG. In contrast, *TNC* was downregulated in both cell lines. However, it exhibited an ~2-fold lower expression level in CAβ3 cells (FC = −3.59) compared to WTβ3 cells (FC = −2.25). Relative to the organization and remodeling of the ECM in the TM and in glaucoma [60], we observed a large decrease in *MMP1* expression in both cell lines (CAβ3 FC = −6.35, WTβ3 FC = −3.59). *TIMP3* was also downregulated in both cell lines (CAβ3 FC = −2.02, WTβ3 FC = −1.57). 

### 3.5. Hierarchical Clustering Reveals Differences in the Expression of Genes Associated with Cell Adhesion

In addition to the differential expression of ECM genes, multiple genes for proteins within the cadherin (CAD) family of cell adhesion receptors met the ±1.5 Log_2_ FC cutoff. A total of 16 genes in this category were identified in either the WTβ3 or CAβ3 cell lines along with other genes associated with cell adhesion (Figure 6). Among those genes were two classical cadherin genes, *CDH6* (K-cadherin) and *CDH11*. *CDH6* was uniquely upregulated in CAβ3 cells (FC = 1.97) while *CDH11* was uniquely downregulated in WTβ3 cells (FC = −3.12). *CDH6* is downregulated in bovine TM cells in response to dexamethasone treatment [77], and both *CDH6* and *CDH11* have been suggested to play a role in regulating IOP (Table 3).

Among the CAD-associated receptors, 14 members of the protocadherin family which mediate homophilic cell-cell adhesions were identified. Of these, 6/14 were uniquely altered in CAβ3 cells: *DCHS1* (*PCDH16*; FC = 3.65), *PCDH1* (FC = 2.47), *PCDHB11* (FC = 2.09), *PCDHB14* (FC = 1.86), *PCDH18* (FC = 2.02) and *PCDHGA6* (FC = −1.60). In contrast 3/14 were uniquely altered in WTβ3 cells: *PCDHGB3* (FC = 1.73), *PCDHGB1* (FC = 1.59) and *PCDH20* (FC = −1.63). The remaining 5/14 (*PCDHB5*, *PCDHB10, PCDHB12*, *PCDHB13* and *PCDHB16*) were upregulated in both cell lines (Appendix A) to a similar extent except for *PCDHB10* which was expressed at a higher level in CAβ3 cells (FC = 3.39) than in WTβ3 cells (FC = 1.93). The significance of these genes in the pathogenesis of POAG is unclear since the functions of many of these protocadherins are still being unraveled.

Other classes of adhesion molecules demonstrated differentially expressed genes besides members of the CAD family including members of the semaphorin family, the IgG superfamily (IGSF), the TIE receptor family, a CD34 family member and an axon guidance molecule. Of interest is *SEMA6A* (FC = 1.55) which was uniquely upregulated only in CAβ3 cells and is also found on the glaucoma locus GLC1G (Table 4). *NCAM2* is also of interest as it has been found to be associated with POAG (Table 2) and is uniquely upregulated in CAβ3 cells (FC = 1.78). *RGMA* encodes a member of the repulsive guidance molecule family that regulates axon guidance and can act as a co-receptor for TGFβ signaling [78]. It is only upregulated in CAβ3 cells (FC = 2.16), and elevated *RGMA* expression has been associated with glaucoma in mice [79].

Several genes that regulate adhesion were also found to be differentially expressed in both cell lines. *ANGPT1* is only upregulated in CAβ3 cells (FC = 1.76), and, in addition to having a role in Schlemm canal development [82], it has been mapped to the CLC1D glaucoma locus (Table 4). *CNTN3* (FC = −3.17), *MSLN* (FC = −1.60), *CD274* (FC = −1.83), *PARVB* (FC = −2.55) and *BMPER* (FC = −2.07) were all downregulated in CAβ3 cells. Except for *MSLN* (Table 2), the proteins encoded by these genes have not been reported to be associated with glaucoma. The BMPER protein, however, has been suggested to play a role in TM function under both normal and glaucomatous conditions as it antagonizes BMP signaling which, in turn, promotes TGFβ2 signaling (Table 3).

### 3.6. Hierarchical Clustering Reveals Differences in Genes Encoding Cytokines, Cytokine Regulators and Genes Involved in TGFβ, BMP and WNT Signaling

Another group of genes whose expression was significantly affected by overexpression or activation of αvβ3 was cytokines, cytokine receptors and cytokine-associated genes (Figure 7). Since it is well established that TGFβ signaling plays a significant role in both glaucoma and IOP regulation [2] it was noteworthy that hierarchical clustering revealed that variable differences in the expression of genes associated with TGFβ signaling were observed between the two cell lines (see WTβ3-CAβ3 column). As shown in Figure 7 (clusters C-E, K, M and P), *TGFB2* (FC = 1.63), *BMP3* (FC = 3.82), *BMP4* (FC = 1.69) and *SMAD6* (FC = 2.19) were all expressed at higher levels in CAβ3 cells relative to WTβ3 cells. In contrast, other TGFβ signaling genes exhibited decreased expression. *INHBB* (FC = −1.65) and *GREM1* (FC = −1.965) were downregulated in CAβ3 cells while *GDNF* (FC = −2.19) was downregulated in WTβ3 cells. Finally, *GDF15* (CAβ3 FC = −2.69; WTβ3 FC = −1.73) was downregulated in both cell lines.

Hierarchical clustering also showed several Wnt-related genes differentially expressed in the two cell lines. (Figure 7, clusters F, J). *WNT8B* was uniquely upregulated in CAβ3 cells (FC = 1.90) while *WNT7B* (FC = −1.82) and *DKK1* (FC = −1.91) were downregulated in CAβ3 cells. Although Wnt signaling has been implicated in IOP regulation and glaucoma [64], none of these genes are known to be associated with glaucoma. The gene *DCDC2* (cluster F) was also uniquely upregulated in CAβ3 cells (FC = 2.025). Although not a member of the Wnt family, the DCDC2 protein has been reported to inhibit β-catenin-dependent Wnt signaling [83].

Among the other genes in this group that met the ±1.5 Log_2_ FC cutoff were five members of the CXC chemokine family that were downregulated only in the CAβ3 cell line (Figure 7, clusters L, I, and G). These genes included *CXCL1* (FC = −1.99), *CXCL2* (FC = −2.21) and *CXCL5* (FC = −1.86) which were all expressed at levels ~2.5-fold lower in CAβ3 cells than in WTβ3 cells. *CXCL3* (cluster I, FC = −3.00) and *CXCL8* (cluster G, FC = −2.85) expression levels, in contrast, were 3- and 5-fold lower, respectively, in CAβ3 cells relative to WTβ3 cells.

Several members of the interleukin family (*IL1B, IL11 and IL18*) and one interleukin receptor (*IL31RA*) also met the ±1.5 Log_2_ FC cutoff in CAβ3 cells, but not in WTβ3 cells. *IL18* (cluster I, FC = −3.74) and *IL1B* (cluster H, FC = −3.95) were strongly downregulated in CAβ3 cells at levels 4- and 8-fold lower than what was observed in WTβ3 cells. Interestingly, *IL1B* is induced in immortalized TM cells that express myocilin (MYOC) mutants known to cause POAG [84]. *IL11* (cluster L, FC = −2.05) was ~3-fold lower in CAβ3 cells than in WTβ3 cells and *IL31RA* (cluster G, FC = −3.46) was expressed at ~ 7.5-fold lower levels in CAβ3 cells relative to WTβ3 cells. Only one IL family member gene *IL21R,* met the ±1.5 Log_2_ FC cutoff in WTβ3 cells (cluster P, FC = −1.81). *IL21R* expression was not significantly altered in CAβ3 cells.

Finally, two other genes of interest that appeared on the cytokine heat map were *TNFAIP3* and *PENK*. *TNFAIP3* (Figure 7, cluster K) was downregulated only in CAβ3 cells (FC = −1.92), and changes in expression of this gene have been associated with glaucoma (Table 2). *PENK* (Figure 7, cluster P) which is localized to the GLC1D glaucoma locus (Table 4) was downregulated in both cell lines at comparable levels (CAβ3 FC = −4.23; WTβ3 FC = −4.96). 

### 3.7. Hierarchical Clustering Reveals Differences in Genes Encoding GTP-Binding and Cytoskeleton-Associated Proteins

It is well established that integrin signaling plays a major role in regulating GTPase activity and the cytoskeleton [85]. Additionally, the significance of the Rho GTPase and its downstream effectors in regulating aqueous humor outflow and the pathogenesis of POAG has been well established [86]. Intriguingly many of the genes (15/26) that were identified in this group (Figure 8) are regulated by proteins associated with or regulated by TGFβ and/or BMP signaling (*DIRAS3*, *NHS*, *ARHGAP24*, *MYH15*, *MCF2L*, *RAPGEF4*, *GAS7*, *ARHGAP22*, *SRGAP3*, *IQGAP2*, *SORBS1*, *RUNDC3A*, *AGAP2*, *INA* and *MAP7*) (Appendix A) while 4/26 (*ADAP2*, *RHOD*, *MYBPH* and *S1PR5*) are not. As with the other categories of genes affected by overexpression and/or activation of αvβ3 integrin, hierarchical clustering of these genes that met the ±1.5 Log_2_ FC cutoff revealed differences in gene expression between the WTβ3 and CAβ3 cell lines (Figure 8, clusters A–C and I–L). 

In clusters A and B, five genes were uniquely upregulated in WTβ3 cells relative to EV control cells. In this group we see several members of the Ras GTPase family (*ADAP2*, *DIRAS3* and *RND2*) and proteins that control cytoskeleton remodeling (*NHS*, and *TUBA4A*.) (Appendix A). Of those genes, *DIRAS3* (FC = 1.935) is the only gene with any association with glaucoma or IOP regulation [87].

In contrast to the genes in clusters A and B, the genes in cluster C and D were all downregulated in CAβ3 cells relative to WTβ3 cells. Of the three genes found in cluster C, only *ARHGAP24* (cluster C, FC = −2.04), which encodes a protein that participates in cell-ECM adhesion complexes in cultured HTM cells [88], has any relevance to TM function or glaucoma as it has also been identified in cultured ONH astrocytes as being phosphorylated in response to increases in hydrostatic pressure [89]. *LPXN* (FC = −1.51; cluster D) encodes a focal adhesion protein that can also function as a transcription factor [90]. It is a member of the paxillin family along with HIC5 which has been shown to play a role in TGF-β2-induced fibrogenic activity and dexamethasone-induced MYOC expression in HTM cells [91].

All six genes in clusters I-L were upregulated only in CAβ3 cells. Of these genes, however, only *AGAP2* (FC = 1.77) is likely to play an important role in regulating TM function as it relates to IOP and POAG pathogenesis. *AGAP2* is a TGFβ-responsive gene that encodes a positive regulator of TGFβ signaling as it relates to fibrosis [92].

### 3.8. Hierarchical Clustering of Miscellaneous and Unclassified Genes

Hierarchical clustering was also performed on the remaining genes that were identified using the ±1.5 Log_2_ FC cutoff (Figure 9). As with the other hierarchical clustering maps, we observed genes that exhibited clear differences in expression between WTβ3 and CAβ3 cells (clusters A, H, I, J, M and O). Of the genes found within these clusters, however, only five have any reported association with IOP regulation or glaucoma. In cluster O, *CYP1B1* was uniquely downregulated in WTβ3 cells (FC = −2.28). *CYP1B1* is associated with glaucoma (Table 2), and mutations in this gene in mice are associated with increases in IOP, oxidative stress and TM dysgenesis [55]. Four other genes worth noting, *TXNIP*, *CA9, RBP7* and *MBP* are found within clusters H, I and J, respectively. *TXNIP* expression was uniquely upregulated in CAβ3 cells (FC = 3.64). It is part of the thioredoxin (TRX) system that protects cells from damage due to oxidative stress and functions in suppressing EMT induced by high glucose [93] or TGFβ (Appendix A). Its expression has not been studied in the TM, but *TXNIP* is upregulated in retinal ganglion cells in response to experimentally induced increases in IOP [94]. *CA9* (FC = 2.43), which was upregulated only in WTβ3 cells, encodes for carbonic anhydrase 9 which appears to play a role in IOP regulation rather than in the pathogenesis of POAG [95]. *RBP7* which is located on the glaucoma locus GLC3B (Table 4) was uniquely upregulated in WTβ3 cells (Log_2_ FC = 1.925). *MBP* was uniquely downregulated in CAβ3 cells (FC = −2.62). Elevated MBP serum levels as well as autoantibodies against the protein have been reported in patients with POAG [96].

### 3.9. RT-qPCR Validation of Changes in Gene Expression 

We then used RT-qPCR to validate selected genes that were altered in these categories. Table 5 shows the primer sequences used for the analysis. Overall, the data obtained through RNA-Seq analysis and RT-qPCR analysis were consistent with each other. As shown in Figure 10, RT-qPCR showed that *TGFB2* and *ADCY1* were both expressed at significantly higher levels in both WTβ3 (*TGFB2*, *p* < 0.0035; *ADCY1*, *p* < 0.002) and CAβ3 cells (*TGFB2*, *p* < 0.0015; *ADCY1*, *p* < 0.002) relative to control EV cells. Analysis of expression of *CDH6* (*p* < 0.015), *SPOCK1* (*p* < 0.002) and *PAX6* (*p* < 0.0015) found that all three genes were significantly upregulated in CAβ3 cells relative to both EV and WTβ3 cells as predicted by the RNA-Seq studies. As predicted by RNA-Seq analysis, *GREM1* (both, *p* < 0.0015)*, MMP1* (both, *p* < 0.002)*, DKK1* (both, *p* < 0.0015), *BMPER* (WTβ3, *p* < 0.002, CAβ3, *p* < 0.0015), *DDIT3* (both, *p* < 0.0011) and *PENK* (both, *p* < 0.0012) were all downregulated in both cell lines compared to EV cells. Finally, RT-qPCR found *VEGFA* expression to be significantly downregulated in CAβ3 cells relative to both EV (*p* < 0.0015) and WTβ3 cells (*p* < 0.002) which is consistent with RNA-Seq analysis. However, by RT-qPCR there was no significant difference in *VEGFA* expression between EV and WTβ3 even though the RNA-Seq analysis found a statistically significant downregulation of this gene in WTβ3 cells relative to EV cells (FC = −0.325, *p* < 0.006).

We also sought to determine if, in addition to the upregulation of TGFB2 mRNA expression observed in CAβ3 cells relative to EV cells, there was increased production of TGFβ2 protein by CAβ3 cells. As shown in Figure 11, ELISA analysis indicated that there was significantly more TGFβ2 in CAβ3 conditioned medium compared to EV conditioned medium (*p* < 0.05). 

### 3.10. Panther Pathway Analysis Finds Gene Overrepresentation Only in CAβ3 Cells

We next determined which signaling pathways were enriched with changes in gene expression. For each cell line, genes identified as being upregulated or downregulated using the Log_2_ ± 1.5 FC cutoff (Figure 2, Appendix A) were entered into the PANTHER pathway database. Although no pathways were found to be statistically overrepresented in WTβ3 cells, three pathways were significantly (*p* < 0.05) overrepresented in CAβ3 cells (Table 6). The pathways overrepresented in CAβ3 cells are involved in TGFβ2, Wnt and cadherin signaling. This was a striking finding given the roles that each of these pathways, either individually or combined, appear to play in both normal IOP regulation and POAG pathogenesis [64]. These results support the idea that it is the activation, rather than simple overexpression, of the αvβ3 integrin that is important in significantly altering gene expression in TM cells.

Since all three of these pathways have been implicated in IOP regulation and/or POAG, we examined which of the genes found to be differentially expressed in response to αvβ3 integrin activation could be targeting one or more of these pathways. As shown in Table 7, we identified genes encoding seven transcription factors (*DDIT3*, *ISL1, KLF4*, *KLF2, MYOCD, SOX5* and *STAT5A*), three ECM-associated proteins (*SULF1*, *TLL1* and *TNC*), two GTP-binding or cytoskeleton proteins (*AGAP2*, *LPXN*), two adhesion-associated proteins (*BMPER* and *PCDH1*) and three cytokines (BMP4, *GREM1* and *IL18*) that have been reported to target the TGFβ2, BMP and/or Wnt signaling pathways. Additionally, we identified many genes from these classes that were targets for these same signaling pathways (Appendix A)

### 3.11. PPI Network Analysis Shows Enrichment Only in CAβ3 Cells

To further examine the relationship between the proteins expressed by the genes that met the ±1.5 Log_2_ FC cutoff for each cell line, we submitted the gene lists from each cell line for analysis into the STRING database in order to generate protein-protein interaction (PPI) networks. Input lists consisted of the uniquely altered genes from each cell line as well as genes that were shared by both cell lines. Surprisingly, analysis of the PPI networks generated using the genes from the WTβ3 cells found only a modest number of interactions between the genes (Figure 12A,B) and statistical analysis of the interactions at two different confidence levels (*p* = 0.143, medium confidence; *p* = 0.259, high confidence) found that these groups of interactions were no different from those of a random collection of input genes. Thus, the limited number of potential interactions in WTβ3 cells was unlikely to result in significant changes to the signaling network in these cells. In contrast, analysis of the networks calculated using the input genes from CAβ3 cells showed many possible interactions at both confidence levels (Figure 13 and Figure 14). Statistical analysis of the CAβ3 networks at both medium and high confidence levels resulted in PPI enrichment *p*-values of 1 × 10^−16^ indicating that the interactions of the gene products from the CAβ3 cells were significantly more than those expected for a random collection of input genes. This suggests that activation of αvβ3 integrin results in a high number of both physical and functional interactions between the proteins expressed by the input genes that could result in significant changes to the signaling network in these cells. 

We then took the list of altered genes in CAβ3 cells that PANTHER identified as being overrepresented for the TGFβ2, Wnt and Cadherin pathways and generated medium and high confidence PPI networks (Figure 15A,B). In addition to the genes identified by PANTHER, we included four genes that demonstrated significantly altered expression in CAβ3 cells, *DKK1* [114], *BMPER* [70], *RGMA* [78] and *GREM1* [69], that have also been connected to one or more of these three signaling pathways. Despite consisting of only 31 genes in total, the medium and high confidence networks that were generated had PPI enrichment *p*-values of 1 × 10^−16^ and 5.72 × 10^−12^, respectively, suggesting that these networks could be significantly altered in CAβ3 cells. 

## 4. Discussion

In this paper we show that activation of αvβ3 integrin signaling in trabecular meshwork cells causes a differential expression of multiple genes that contribute to TGFβ2 signaling and fibrotic-like changes that could lead to the development of glaucoma [2]. Focusing on genes that had altered expression of ± 1.5 Log_2_ fold change (FC), we found that overexpressing a CA-αvβ3 integrin resulted in the differential expression of genes compared to cells overexpressing a WT-αvβ3 integrin. In particular, we found an enrichment for genes encoding proteins found in the TGFβ2, Wnt and cadherin signaling pathways all of which have been implicated in the pathogenesis of POAG [2,64]. Among the significant changes detected was an upregulation in *TGFB2* which is found to be elevated in 50% of patients with POAG. *CDH6*, *BMP4*, *MMP1* and *TIMP3* are other genes found to be differentially expressed and associated with TGFβ2 signaling, remodeling of the TM ECM and/or increased IOP (Table 2). In contrast, no enrichment for genes in any specific signaling pathway was detected in TM-1 cells overexpressing WT-αvβ3 integrin. Many of the differentially expressed genes detected belong to several classes that may contribute to the development of POAG. These include genes encoding transcription factors, adhesion and cytoskeleton proteins, ECM proteins, cytokines and GTPases. These findings, together with our earlier studies [4,6,11,41], support the idea that changes in αvβ3 integrin signaling are likely to contribute to the changes in trabecular meshwork associated with glaucoma. 

One of the most striking changes observed in this study was the increase in *TGFB2* expression along with the expression of genes associated with TGFβ2 signaling. TGFβ2 levels are known to be elevated in the aqueous humor of POAG patients and increased expression of TGFβ2 causes an elevation in IOP in rodent models of ocular hypertension [2]. However, it is not known why the levels are upregulated. The upregulation of *TGFB2* expression in response to αvβ3 integrin signaling in our study would suggest that increases in αvβ3 integrin signaling may be involved. This is not entirely surprising. αvβ3 integrin has been shown to be associated with the upregulation of *TGFB2* in several cell types including a breast cancer epithelial cell line [115] and human skin fibroblasts [116]. 

Recent studies have also highlighted the importance of αvβ3 integrin in TGFβ2-mediated senescence [117] further suggesting that there is a link between αvβ3 integrin activity and TGFβ2 expression. Interestingly, the expression of αvβ3 integrin in human skin fibroblasts and in the liver of C57BL/6J mice correlated with aging [116] and senescence [117]. Elevated αvβ3 integrin levels have also been found to be associated with apoptosis [118]. This raises the interesting question of whether increases in αvβ3 integrin expression in the aged TM could also be contributing to cell loss or senescence in the TM which is known to be associated with POAG and thought to be one of the causes of POAG [119] Notably, Yu, et al. [120] found that TGFβ2 induced senescence-associated changes within HTM cells and activated a senescence-related signaling pathway similar to that reported by Rapisarda, et al. [116]. This suggests an additional mechanism whereby dysregulated αvβ3 integrin signaling together with increased *TGFB2* expression could contribute to POAG pathogenesis.

In addition to *TGFB2*, many of the genes that were altered belonged to classes of proteins such as ECM proteins, cell adhesion molecules and transcription factors, that are known to be altered in glaucoma and either regulate TGFβ2 signaling (Table 7) or are regulated by TGFβ2 signaling (Appendix A). The major gene class that was affected by activation of αvβ3 integrin signaling by far, however, was transcription-associated genes. Several of these genes are known to be associated with glaucoma (Table 2). For instance, *PAX6* is considered a master regulator of TGFβ2 expression in ocular tissues [104]. *FOSL1* [46], *SOX5*, *STAT5A*, *MYOCD*, *ISL1*, *KLF4* and *KLF2* all play a role in fibrosis, myofibroblast differentiation, EMT, TGFβ signaling, BMP signaling, and/or Wnt signaling (Table 7). The significant up and down regulation of these types of transcription factors suggest that αvβ3 integrin signaling is altering multiple signaling pathways involved in the fibrogenic transition of the TM during glaucoma. The significance of these changes is unknown, however, the downregulation of both *KLF2* and *KLF4* expression would potentially attenuate two significant inhibitors of TGFβ2 signaling and thus enhance TGFβ-induced EMT [47,51].

The finding that overexpression of a constitutively active αvβ3 integrin also led to an enrichment in genes associated with TGFβ, Wnt and cadherin signaling pathways further supports the hypothesis that αvβ3 integrin is playing a role in the transcriptional regulation of the biological processes associated with glaucoma. In contrast, overexpression of wild type αvβ3 integrin did not have a significant effect on any of these signaling pathways suggesting that it is the active signaling conformation of αvβ3 integrin which is more important than its expression levels or adhesion to the ECM. 

It is noteworthy that several other genes that were upregulated by αvβ3 integrin signaling are associated with various types of glaucoma and IOP regulation (Table 2, Table 3 and Table 4) further supporting the idea that αvβ3 integrin and the concomitant increase in *TGFB2* expression together may be playing a role in glaucoma. Among the genes upregulated and associated with glaucoma were several components of the ECM (*POSTN*, *COL8A2*, *SEMA6A*) as well as *ANGPT1* and *BMP4*. We also found a decrease in expression of several genes associated with the ECM especially *MMP1* and *TIMP3*. These genes all have one thing in common and that is their expression would affect the remodeling of ECM. 

We also saw an effect on genes associated with the development of the anterior segment. For instance, *PAX6* and *LMX1B* are both associated with ASD which, in turn, is associated with an increased risk of glaucoma [53,54,121]. Although it does not encode a transcription factor, alterations in *CYP1B1* expression have also been shown to cause ASD [55] and are associated with glaucoma. *LMX1B*, independent of its ASD association, and *GAS7* are also associated with elevated IOP and POAG [54,56,121,122]. Thus, it is interesting that these genes exhibited DGE when αvβ3 integrin is overexpressed and/or activated. This suggests that αvβ3 integrin may play a role in TM development as well as glaucoma.

One surprising finding was the downregulation of several pro-inflammatory cytokines especially several members of the CXC family of chemokines, several members of the interleukin family and VEGFA. Although it is not clear why this would happen, the upregulation of these cytokines is usually associated with acute early inflammatory response [123] and their levels tend to wain during the later fibrotic phase when *TGFB2* is upregulated. Hence, downregulation of chemokine expression along with elevated levels of TGFβ2 could be important determinants in the pathogenesis of fibrosis [124].

Finally, previous studies in our lab have also found that two phenotypes associated with glaucoma, cytoskeletal rearrangements into CLANs and decreased phagocytosis, can be triggered in response to activation of αvβ3 signaling. Both phenotypes are dependent upon signaling pathways regulated by the Rac GTPase [8,9,10,125,126]. Several genes (*ARHGAP22*, *ARHGAP24*, *SRGAP3* and *IQGAP2)* that regulate Rac activity were also differentially expressed in CAβ3 cells and are targets of the TGFβ, BMP and/or Wnt signaling pathways (Appendix A). Whether these genes play a role in CLAN formation or phagocytosis remains to be determined, however, it does further support a role for αvβ3 integrin signaling in the development of these changes in glaucoma. 

In summary, to the best of our knowledge this study is the first to show that increased αvβ3 integrin expression and signaling could be responsible for enhanced TGFβ2 expression in POAG. It also shows that changes in the expression of the normal integrin repertoire on the cell surface of TM cells can have significant effects on the pathophysiology of TM cells in glaucoma and that activation of αvβ3 integrin signaling is not simply involved in mediating adhesion. Rather, significant downstream signaling events would result from this activation subsequently causing major changes in the TM transcriptome and TM function. Hence, these studies, together with our previous studies, show that αvβ3 integrin is an important regulator of the organization of the actin cytoskeleton, phagocytosis and signaling of GTPases such as Rac in TM cells thus suggesting that alterations in cell-matrix signaling are likely to contribute to the dysregulation of trabecular meshwork function that leads to glaucoma. 

## Figures and Tables

**Figure 1 cells-10-01923-f001:**
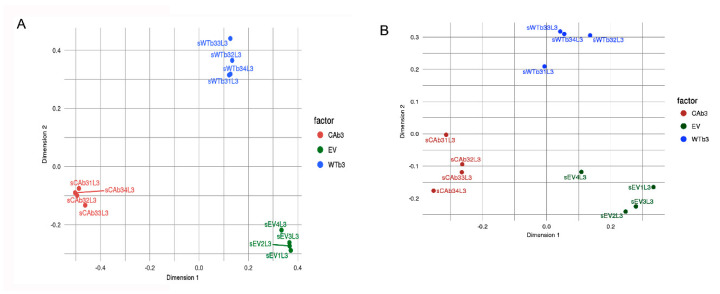
The EV, WTβ3-integrin and CAβ3-integrin cell lines are distinct cell populations with unique transcriptomes. Unsupervised clustering was used to verify that the cell lines used were distinct from each other. Multidimensional scaling (MDS) plots at the gene (**A**) and the transcript (**B**) level were generated. Both plots show that the greatest variation observed was between the three cell lines rather than within the individual samples of each cell line.

**Figure 2 cells-10-01923-f002:**
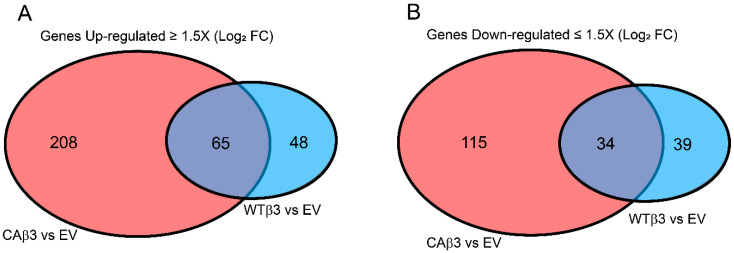
CAβ3 and WTβ3 cells have unique and shared sets of genes with altered expression. Venn diagram analysis was performed looking for differentially expressed genes that demonstrated a Log_2_ FC ± 1.5 in CAβ3 or WTβ3 cells relative to control EV cells. The datasets from the CAβ3-EV and WTβ3-EV contrasts were analyzed using R statistical computing and graphics software. Shared and unique sets of genes that were upregulated (**A**) and downregulated (**B**) relative to EV cells were identified. The complete lists of unique and shared genes that met the FC cutoff are found in Appendix A.

**Figure 3 cells-10-01923-f003:**
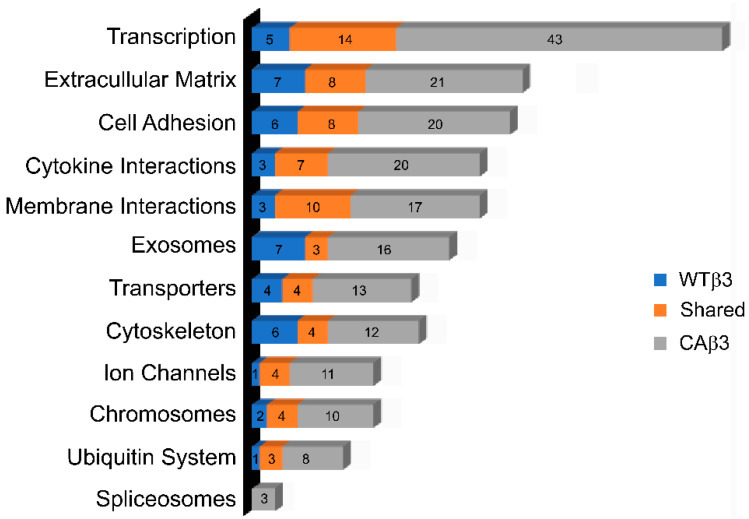
More biological processes were affected by overexpressing CA-αvβ3 integrin than WT-αvβ3 integrin. Biological processes that were enriched in CAβ3 or WTβ3 cells were identified through gene ontology enrichment analysis and gene classification that was performed on the genes that met the Log_2_ FC ± 1.5 cutoff from each cell line using a combination of the following databases: KEGG, DAVID Bioinformatics Resources, the Matrisome Project and UniProt. The top twelve processes are highlighted. The remaining processes that were affected can be found in Appendix A.

**Figure 4 cells-10-01923-f004:**
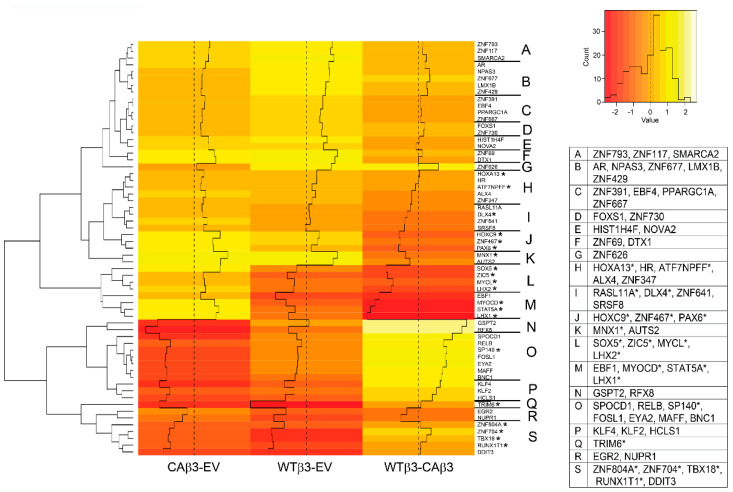
Hierarchical clustering of transcription-associated genes finds differential gene expression in WTβ3 and CAβ3 cells relative to control EV cells and to each other. Genes from each of the three contrasts analyzed for DGE that met the Log_2_ FC ± 1.5 cutoff and were classified as either transcription factors, or otherwise part of the transcription process were analyzed using R statistical computing and graphics software. A heat map displaying standardized z-scores was generated. The clusters identified by letters on the right side of the map were determined manually. Genes within each of these clusters are listed in the displayed table. Those genes highlighted by an asterisk represent genes that had been excluded from the initial DGE analysis due to low abundance in the read count in two or more samples of one of the three cell lines. To be included in the heat map, the missing Log_2_ FC values were calculated from the raw count data using Excel. To be consistent, the corresponding Log_2_ FC values for the same gene in the other two cell lines were re-calculated from the raw data as well. These values were then converted to standardized z-scores along with the other genes using R.

**Figure 5 cells-10-01923-f005:**
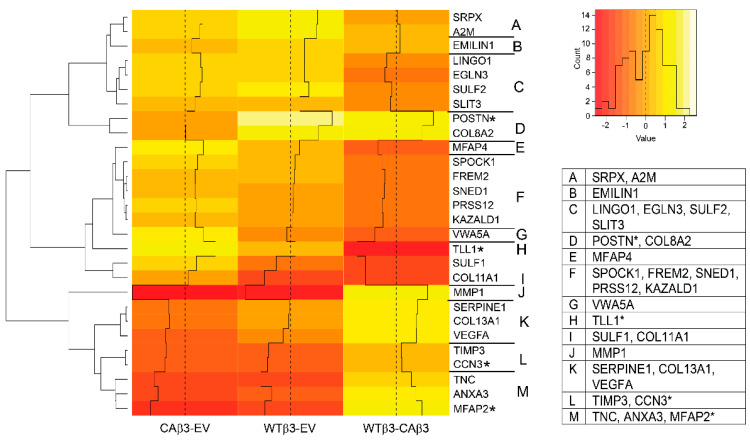
Hierarchical clustering of ECM-associated genes finds differential gene expression in WTβ3 and CAβ3 cells relative to control EV cells and to each other. Genes from each of the three contrasts analyzed for DGE that met the Log_2_ FC ± 1.5 cutoff and were classified as encoding either core ECM proteins, ECM-associated proteins or matricellular proteins were analyzed as in Figure 4. A heat map displaying standardized z-scores was generated. The clusters identified by letters on the right side of the map were determined manually. Genes within each of these clusters are listed in the displayed table. Those genes highlighted by an asterisk represent genes that had been dealt with as described for Figure 4.

**Figure 6 cells-10-01923-f006:**
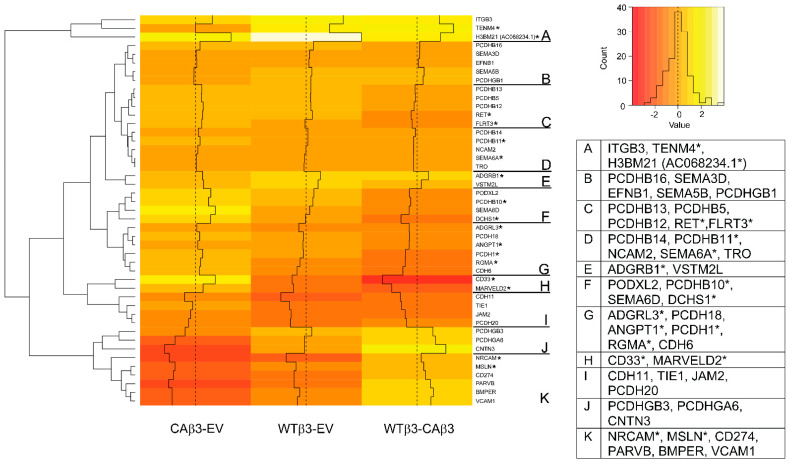
Hierarchical clustering of adhesion-associated genes finds differential gene expression in WTβ3 and CAβ3 cells relative to EV control cells and to each other. Genes from each of the three contrasts analyzed for DGE that met the Log_2_ FC ± 1.5 cutoff and were classified as encoding either adhesion proteins or adhesion-associated proteins were analyzed as in Figure 4. A heat map displaying standardized z-scores was generated. The clusters identified by letters on the right side of the map were determined manually. Genes within each of these clusters are listed in the displayed table. Those genes highlighted by an asterisk represent genes that had been dealt with as described for Figure 4.

**Figure 7 cells-10-01923-f007:**
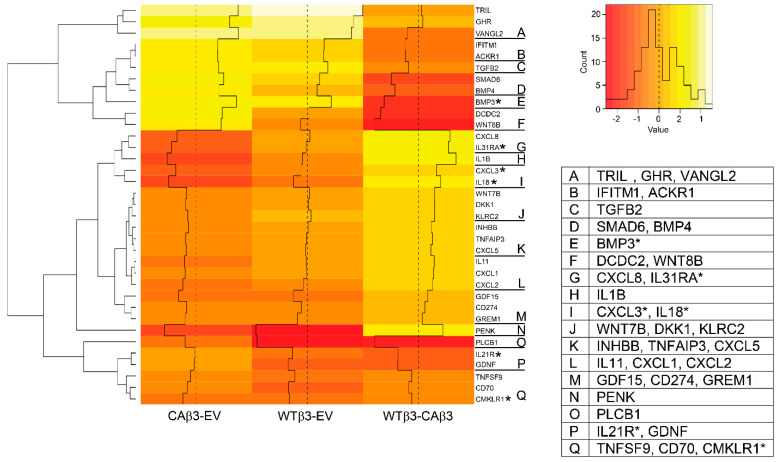
Hierarchical clustering of cytokines and cytokine-associated genes finds differential gene expression in WTβ3 and CAβ3 cells relative to control EV cells and to each other. Genes from each of the three contrasts analyzed for DGE that met the Log_2_ FC ± 1.5 cutoff and were classified as encoding either cytokines or cytokine-associated proteins were analyzed as in Figure 4. A heat map displaying standardized z-scores was generated. The clusters identified by letters on the right side of the map were determined manually. Genes within each of these clusters are listed in the displayed table. Those genes highlighted by an asterisk represent genes that had been dealt with as described for Figure 4.

**Figure 8 cells-10-01923-f008:**
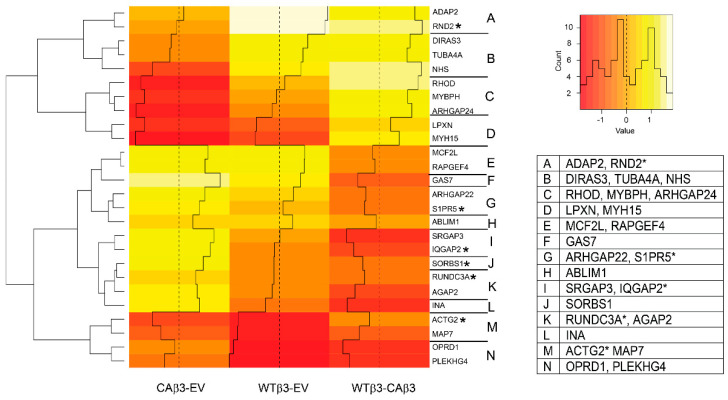
Hierarchical clustering of GTP-binding and cytoskeleton-associated genes finds differential gene expression in WTβ3 and CAβ3 cells relative to control EV cells and to each other. Genes from each of the three contrasts analyzed for DGE that met the Log_2_ FC ± 1.5 cutoff and were classified as encoding GTP-binding and/or cytoskeleton-associated proteins were analyzed as in Figure 4. A heat map displaying standardized z-scores was generated. The clusters identified by letters on the right side of the map were determined manually. Genes within each of these clusters are listed in the displayed table. Those genes highlighted by an asterisk represent genes that had been dealt with as described for Figure 4.

**Figure 9 cells-10-01923-f009:**
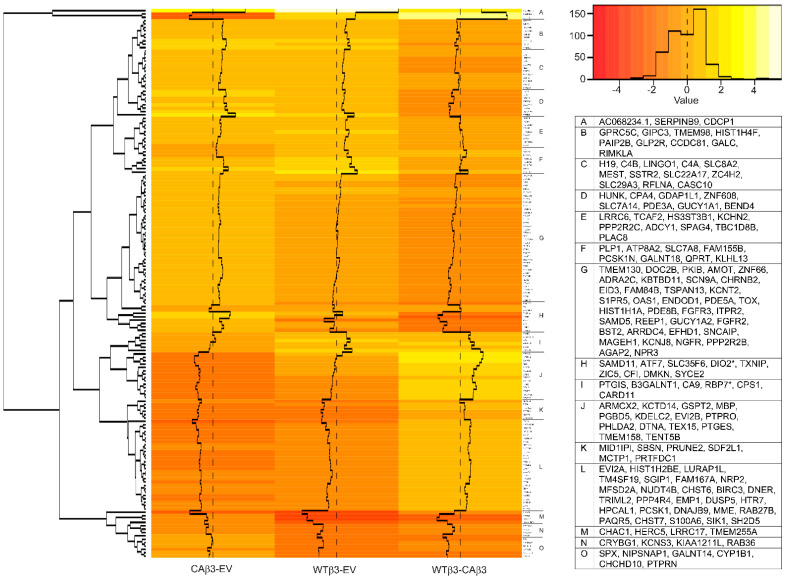
Hierarchical clustering of miscellaneous and unclassified genes finds differential gene expression in WTβ3 and CAβ3 cells relative to control EV cells and to each other. Miscellaneous and unclassified genes from each of the three contrasts analyzed for DGE that met the Log_2_ FC ± 1.5 cutoff were analyzed as in Figure 4. A heat map displaying standardized z-scores was generated. The clusters identified by letters on the right side of the map were determined manually. Genes within each of these clusters are listed in the displayed table. Those genes highlighted by an asterisk represent genes that had been dealt with as described for Figure 4.

**Figure 10 cells-10-01923-f010:**
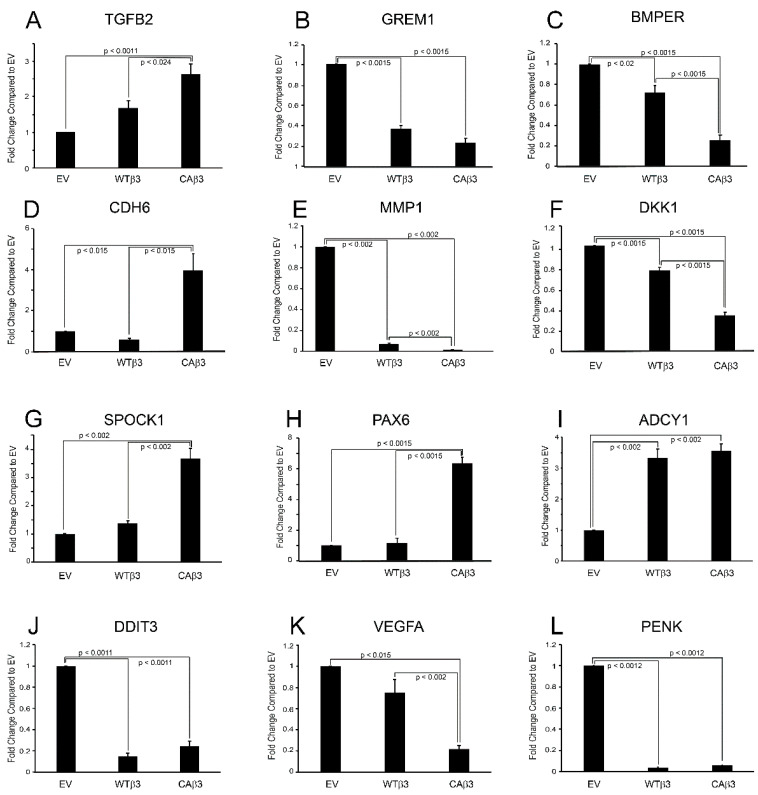
Real-time (RT)-qPCR validates RNA-Seq data. Expression levels of twelve genes selected from the various categories were analyzed using RT-qPCR. Primers used are shown in Table 5. Expression levels for each gene were compared to the housekeeping gene *SDHA*. For each gene, samples were analyzed in triplicate (**B**–**D**,**F**,**H**) or quadruplicate (**A**,**E**,**G**,**I**–**L**). Results are reported as the average fold change ± SEM, compared to EV.

**Figure 11 cells-10-01923-f011:**
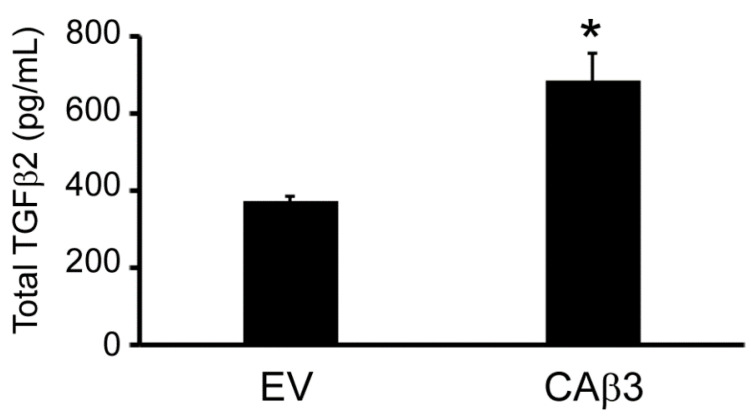
CAβ3 cells secrete more TGFβ2 than EV cells. ELISA data for total TGFβ2 in the medium of EV and CAβ3 cells. TGFβ2 was significantly (* *p* < 0.05) increased in the medium of CAβ3 cells compared to EV cells.

**Figure 12 cells-10-01923-f012:**
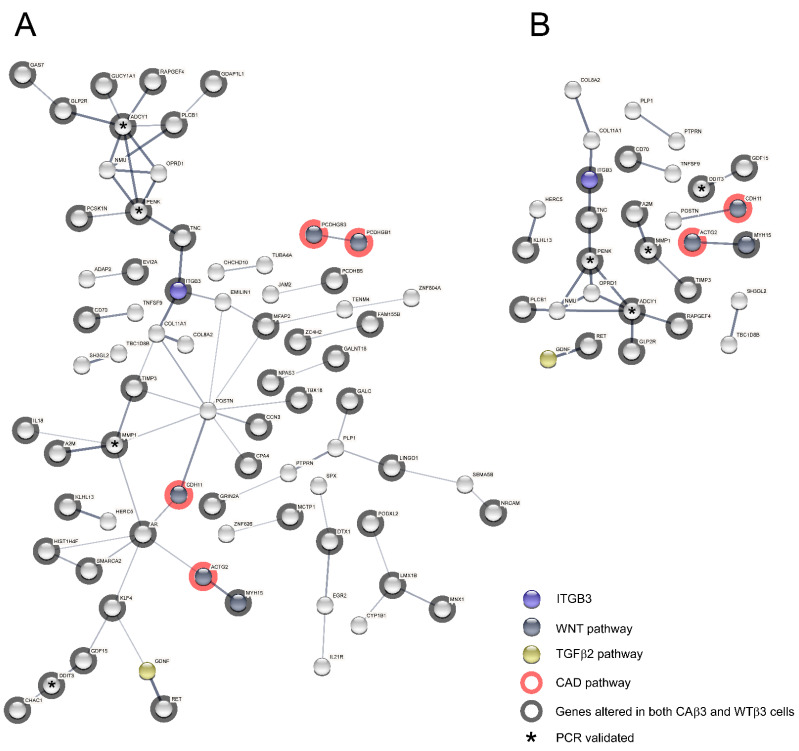
WTβ3 protein-protein interaction (PPI) networks are similar to those of a random collection of proteins. Differentially expressed genes resulting from overexpression of a WT-αvβ3 integrin were entered into the STRING database. The networks display the proteins encoded by each input gene. The input consisted of uniquely altered genes as well as altered genes that were shared with cells overexpressing a CA-αvβ3 integrin. (**A**) Network calculated using a medium (0.4) statistical confidence; PPI enrichment *p*-value = 0.143. (Interactive link for Figure 12A: permanent link: https://version-11-0b.string-db.org/cgi/network?networkId=bEnUFrdwbGJv) (accessed on 20 June 2021) (**B**) Network calculated using a high (0.7) statistical confidence; PPI enrichment *p*-value = 0.259. (Interactive link for Figure 12B: permanent link: https://version-11-0b.string-db.org/cgi/network?networkId=b9wf05GlAE9T) (accessed on 20 June 2021). The confidence score is the approximate probability that a predicted link exists between two proteins in the same metabolic map in the KEGG database. Both PPI enrichment scores indicate that the calculated networks are not significantly different from networks generated by a random collection of input genes at the same confidence level. For both networks, the proteins for which there were no connections to be mapped were not displayed. The location of β3 integrin (INTB3), members of the TGFβ2, Wnt and Cadherin signaling pathways as well as those gene products that are also differentially expressed in CAβ3 cells are all indicated. * Genes that were validated by RT-qPCR.

**Figure 13 cells-10-01923-f013:**
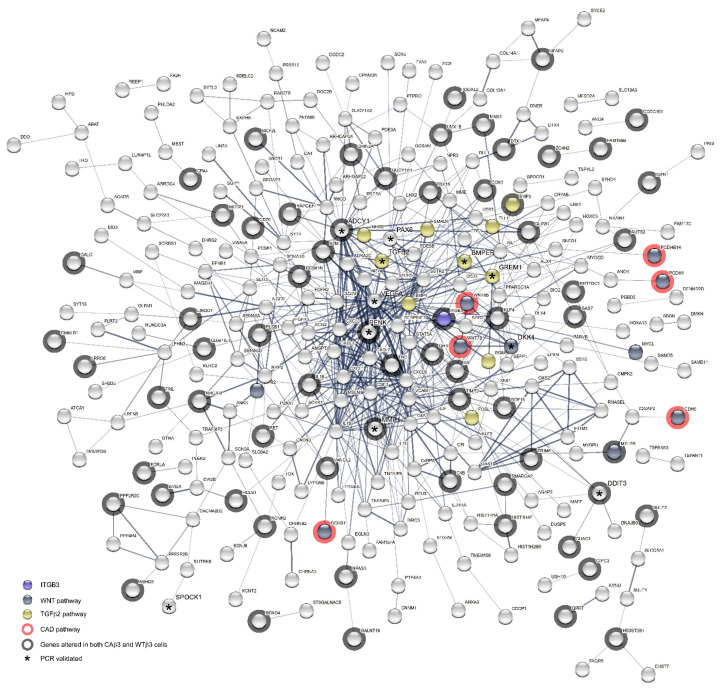
Medium confidence CAβ3 PPI network shows an extensive network of potential interactions. Differentially expressed genes resulting from overexpression of a CA-αvβ3 integrin were entered into the STRING database. The input consisted of uniquely altered genes as well as altered genes that were shared with cells overexpressing a WT-αvβ3 integrin. The calculated network (statistical confidence = 0.4) has a PPI enrichment value = 1 × 10^−16^ indicating that the network has significantly more interactions than expected for a random collection of input genes at the same confidence level. The location of β3 integrin (INTB3), members of the TGFβ2, Wnt and Cadherin signaling pathways as well as those gene products that are also differentially expressed in CAβ3 cells are all indicated. * Genes that were validated by RT-qPCR. The proteins for which there were no connections to be mapped where not displayed. (Interactive link for Figure 13: permanent link: https://version-11-0b.string-db.org/cgi/network?networkId=bfy9gG5S77FU) (accessed on 20 June 2021).

**Figure 14 cells-10-01923-f014:**
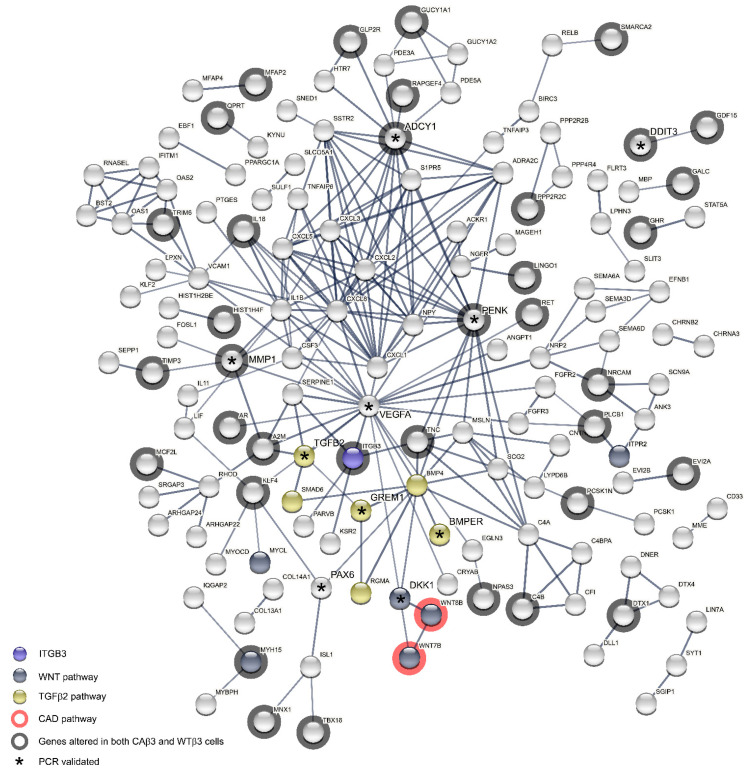
High confidence CAβ3 PPI network shows an extensive network of potential interactions. Differentially expressed genes resulting from overexpression of a CA-αvβ3 integrin were entered into the STRING database. The input consisted of uniquely altered genes as well as altered genes that were shared with cells overexpressing a WT-αvβ3 integrin. The calculated network (statistical confidence = 0.7) has a PPI enrichment value = 1 × 10^−16^ indicating that the network has significantly more interactions than expected for a random collection of input genes at the same confidence level. The location of β3 integrin (INTB3), members of the TGFβ2, Wnt and Cadherin (CAD) signaling pathways as well as those gene products that are also differentially expressed in WTβ3 cells are all indicated. * Genes that were validated by RT-qPCR. The proteins for which there were no connections to be mapped where not displayed. (Interactive link for Figure 14: permanent link: https://version-11-0b.string-db.org/cgi/network?networkId=bZECX4AuPQ32) (accessed on 20 June 2021).

**Figure 15 cells-10-01923-f015:**
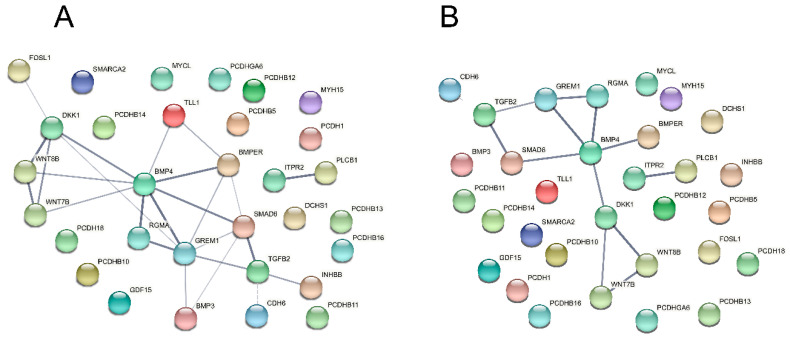
PPI networks consisting only of differentially expressed genes from CAβ3 cells that were overrepresented in the TGFβ2, Wnt and CAD signaling pathways demonstrate a significant number of potential interactions. Genes from Table 6 along with four additional genes (*DKK1*, *BMPER*, *RGMA* and *GREM1*) that have been shown to participate in one or more of these pathways, were entered into the STRING database. (**A**) Network calculated using a medium (0.4) statistical confidence; PPI enrichment *p*-value = 1 × 10^−16^. (Interactive link for Figure 15A: permanent link: https://version-11-0b.string-db.org/cgi/network?networkId=bxHIv7XuhKtR) (accessed on 20 June 2021). (**B**) Network calculated using a high (0.7) statistical confidence; PPI enrichment *p*-value = 5.72 × 10^−12^. (Interactive link for Figure 15A: permanent link: https://version-11-0b.string-db.org/cgi/network?networkId=bWKleDuHyVqt) (accessed on 20 June 2021). Both PPI enrichment scores indicate that these networks have significantly more interactions than expected for a random collection of input genes at the same confidence level. The proteins for which there were no connections to be mapped are also displayed.

**Table 1 cells-10-01923-t001:** Summary of differential gene expression analysis for TM-1 cells overexpressing either a wild type β3 integrin (WTβ3) or a constitutively active β3 integrin (CAβ3). Control cells were transduced with an empty vector (EV). These data can be accessed in the Appendix A. The complete data set has also been deposited in NCBI’s Gene Expression Omnibus and is accessible through GEO Series. Accession number: GSE180407. # = number.

Contrast	Total # of Genes with Altered Expression Levels (*p* < 0.05)	# of Upregulated Genes	# of Downregulated Genes
WTβ3-EV	5186	2645	2541
CAβ3-EV	8249	4094	4255
WTβ3-CAβ3	6999	3524	3475

**Table 2 cells-10-01923-t002:** Differentially expressed genes in WTβ3 and/or CAβ3 cells that are associated with glaucoma. Except where noted by *, Log_2_ FC values were reported in DGE analysis using EdgeR. * Genes that were filtered out due to low abundance did not originally have a FC determined (ND). Log_2_ FC was calculated from the raw count data. The values in parentheses were originally calculated using EdgeR.

Glaucoma Association	Gene Symbol	Protein Name	Protein Function	Log_2_ FC WTβ3 Cells	Log_2_ FC Caβ3 Cells	References
ASD						
	*PAX6*	Paired Box 6	Transcription factor	0.43 * (ND)	2.97 * (2.78)	[53,54]
	*CYP1B1*	Cytochrome P450 Family 1 Subfamily B Member 1	Monooxygenase	−2.28	−0.37	[53,55]
	*POSTN*	Periostin	Supports epithelial cell adhesion and migration	3.88	0.53	[55]
NTG						
	*GAS7*	Growth Arrest Specific 7	DNA-binding transcription factor activity and actin filament binding	3.21	1.5	[54,56]
PACG						
	*COL11A1*	Collagen Type XI Alpha 1 Chain	Extracellular Matrix protein	0.75	−1.95	[57]
POAG						
	*TGFB2*	Transforming Growth β2	Growth factor/Cytokine	1.62	1.23	[2]
	*LMX1B*	LIM Homeobox Transcription Factor 1 Beta	Transcription factor	1.65	1.52	[58,59]
	*SPOCK1*	Testican/Osteonectin	Matricellular protein; induces EMT	1.92	0.70	[60,61]
	*MSLN*	Mesothelin	Cytokine	−0.73	−1.60	[54]
	*TIMP3*	TIMP Metallopeptidase Inhibitor 3	Metalloproteinase inhibitor	−1.57	−2.02	[60]
	*MMP1*	Matrix Metallopeptidase 1	Metalloproteinase	−6.35	−3.59	[60]
	*GALC*	Galactosylceramidase	Lysosomal enzyme	2.00	3.09	[62]
	*SULF1*	Sulfatase 1	extracellular heparan sulfate endosulfatase	−0.93	1.76	[54]
	*COL8A2*	Collagen Type VIII Alpha 2 Chain	Extracellular Matrix protein	0.41	2.71	[54]
	*COL11A1*	Collagen Type XI Alpha 1 Chain	Extracellular Matrix protein	0.75	−1.95	[63]
	*BMP4*	Bone Morphogenetic Protein 4	Cytokine	1.69	0.30	[60]
	*CDH6*	Cadherin 6	Transmembrane receptor; cell-cell adhesion	−0.59	1.97	[64]
	*NCAM2*	Neural Cell Adhesion Molecule 2	Transmembrane receptor; cell-cell adhesion	0.81	1.78	[65]
	*ABCB1*	ATP binding cassette subfamily B member 1	ATP-dependent drug efflux pump for xenobiotic compounds	0.07	3.5	[66]
	*TNFAIP3*	TNF alpha Induced Protein 3	Ubiquitin-editing enzyme	−0.59	−1.92	[67]

**Table 3 cells-10-01923-t003:** Differentially expressed genes in WTβ3 and/or CAβ3 cells that are associated with IOP regulation. Log_2_ FC values were reported in DGE analysis using EdgeR.

Gene Symbol	Protein Name	Protein Function	Log_2_ FC WTβ3 Cells	Log_2_ FC Caβ3 Cells	References
*TGFB2*	TGFβ2	Growth factor/cytokine	1.23	1.63	[2]
*CHD6*	Cadherin 6	Cell adhesion	−0.59	1.97	[64]
*CDH11*	Cadherin 11	Cell adhesion	−3.12	−0.19	[68]
*GREM1*	gremlin	BMP inhibitor	−1.38	−1.965	[69]
*BMPER*	bumper	BMP inhibitor	−0.155	−2.07	[70]
*DKK1*	Dickkopf WNT Signaling Pathway Inhibitor 1	Inhibits beta-catenin-dependent Wnt signaling	−0.44	−1.91	[71]
*DDIT3*	DNA Damage Inducible Transcript 3	Transcription factor	−2.90	−2.25	[72]

**Table 4 cells-10-01923-t004:** Differentially expressed genes in WTβ3 and/or CAβ3 cells that are located on a glaucoma locus. Except where noted by *, Log_2_ FC values were reported in DGE analysis using EdgeR. * Genes that were filtered out due to low abundance did not originally have a FC determined (ND). Log_2_ FC was calculated from the raw count data. The values in parentheses were originally calculated using EdgeR.

TM Function	Gene Symbol	Protein Name	Locus	Protein Function	Log_2_ FC WTβ3 Cells	Log_2_ FC Caβ3 Cells	References
SC Devel.							
	*ANGPT1*	Angiopoietin 1	GLC1D	Secreted glycoprotein; TEK/TIE2 ligand	0.365 * (ND)	2.07 * (1.75)	[80]
PEX							
	*SEMA6A*	Semaphorin 6A	GLC1G	Transmembrane receptor; cell-cell adhesion	0.56	1.55	[80]
UNK							
	*PENK*	Proenkephalin	GLC1D	Opioid neuropeptide	−4.95	−4.22	[80,81]
	*DIO2*	Iodothyronine Deiodinase 2	GLC3G	Oxidoreductase activity	1.35 * (ND)	3.98 * (3.77)	[80]
	*RBP7*	Retinol Binding Protein 7	GLC3B	Retinol-binding protein	1.96 * (1.925)	0.11 * (ND)	[80]

**Table 5 cells-10-01923-t005:** Primer sequences used for RT-qPCR validation of select genes identified by RNA-Seq analysis. All sequences are for the human version of each gene.

Type	Sequence
*TGFB2*	f- CAGCACACTCGATATGGACCA r- CCTCGGGCTCAGGATAGTCT
*DDIT3*	f- TTGCCTTTCTCCTTCGGGAC r- CAGTCAGCCAAGCCAGAGAA
*CDH6*	f- AGAACTTACCGCTACTTCTTGC r- TGCCCACATACTGATAATCGGA
*BMPER*	f- GTGCTTGTGTGAAAGGCAGG r- AAACGTACTGACACGTCCCC
*GREM1*	f- ACTCTCGGTCCCGCTGA r- CAAGAGGAGAAGCAGGGCTC
*MMP1*	f- GGTGTGGTGTCTCACAGCTT r- CGCTTTTCAACTTGCCTCCC
*SPOCK1*	f- GACCTCCTGCTTGACCCTTC r- TTTGTCCACACACCAGCACT
*PAX6*	f- TCAAGCAACAACAGCAGCAC r- TCACTCCGCTGTGACTGTTC
*ADCY1*	f- GTCGGATGGATAGCACAGGG r- TTTGGGAGCCGTTTCCATCA
*PENK*	f- ATCCTCGCCAAGCGGTATG r- GGTTGTCCCCTGTTTCCAGA
*VEGFA*	f- CTGCTGTCTTGGCTGCATTGG r- CACCGCCTCGGCTTGTCACAT
*DKK1*	f- GTGCAAATCTGTCTCGCCTG r- GCACAGTCTGATGACCGGAG
*SDHA*	f- TGGGAACAAGAGGGCATCTG r- CCACCACTGCATCCAATTCATG

**Table 6 cells-10-01923-t006:** Pathways enriched by activation of αvβ3 integrin compared to control cells. Genes identified as being upregulated or downregulated ± 1.5 Log_2_ FC were entered into the PANTHER database (pantherdb.org) and analyzed using a statistical overrepresentation test (Fisher’s Exact test). A total of 422 genes were entered. 365 genes were successfully mapped to a pathway. 57 genes were left unmapped. The genes associated with each pathway are as follows: TGFβ2 signaling (*TGFB2, BMP3, BMP4, SMAD6, TLL1, GDF15, INHBB, FOSL1*); Wnt signaling (*WNT7B, WNT8B, ITPR2, MYCL, MYH15, PLCB1, SMARCA2, CDH6, DCHS1, PCDH1, PCDH18, PCDHB5, PCDHB10, PCDHB11, PCDHB12, PCDHB13, PCDHB14, PCDHB16, PCDHGA6*) and Cadherin signaling (*CDH6, DCHS1, PCDH1, PCDH18, PCDHB5, PCDHB10, PCDHB11, PCDHB12, PCDHB13, PCDHB14, PCDHB16, PCDHGA6, WNT7B, WNT8B*). # = number.

Pathway	# of Reference Genes	# of Genes	Expected # of Genes	Fold Enrichment	Raw *p*-Value	False Discovery Rate
TGFβ2 signaling	102	8	1.81	4.43	0.000686	0.0286
Wnt signaling	317	19	5.62	3.38	0.0000077	0.000429
Cadherin signaling	164	14	2.91	4.82	0.00000304	0.000254

**Table 7 cells-10-01923-t007:** Differentially expressed genes in CAβ3 cells that target TGFβ2, BMP or Wnt signaling. Log_2_ FC values were reported in DGE analysis using EdgeR.

Functional Classification	Gene Name	Protein Name	CAβ3-EV Log_2_ FC	Effect on TGFβ, BMP or Wnt Signaling
Transcription				
	*DDIT3*	DNA damage inducible transcript 3	−0.2.25	Targets TGFβ; Wnt [97]
	*ISL1*	ISL LIM Homeobox 1	8.81	Targets BMP [98]
	*KLF2*	Kruppel like factor 2	−1.54	Targets TGFβ [99]; Wnt [100]
	*KLF4*	Kruppel like factor 4	−2.59	Targets TGFβ [47]; BMP [99]; Wnt [101]
	*MYOCD*	Myocardin	−2.94	Targets TGFβ [102]; Wnt [103]
	*PAX6*	Paired box 6	2.78	Targets TGFβ [104]
	*SOX5*	SRY-box 5	3.49	Targets Wnt [105]
	*STAT5A*	Signal transducer and activator of transcription 5A	3.32	Targets Wnt [106]
Adhesion				
	*BMPER*	BMP binding endothelial regulator	−2.07	Targets BMP [107]
	*PCDH1*	Protocadherin 1	2.48	Targets TGFβ [108]
ECM				
	*SULF1*	Sulfatase 1	1.76	Targets TGFβ [109]
	*TLL1*	Tolloid like 1	4.19	Targets TGFβ [110]
	*TNC*	Tenascin C	−3.59	Targets TGFβ [111]
Cytokines				
	*BMP4*	Bone morphogenetic protein 4	1.69	Targets TGFβ [69]
	*GREM1*	Gremlin 1	−1.97	Targets BMP [69]
	*IL18*	Interleukin 18	−3.75	Targets TGFβ [112]
Cytoskeleton				
	*AGAP2*	ArfGAP with GTPase domain, ankyrin repeat and PH domain 2	1.77	Targets TGFβ [92]
	*LPXN*	Leupaxin	−1.51	Targets Wnt [113]

## Data Availability

Complete EdgeR datasets for WTb3-EV, CAb3-EV and WTb3-CAb3 contrasts (deposited in NCBI’s Gene Expression Omnibus; accessible through GEO Series. Accession number: GSE180407): Excel file names: edgeRglm_GENE_WTb3-EV, edgeRglm_GENE_CAb3-EV, edgeRglm_GENE_WTb3-CAb3.

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
