# Peer review of "Overexpression and Activation of αvβ3 Integrin Differentially Affects TGFβ2 Signaling in Human Trabecular Meshwork Cells"

_cells, 2021, doi:10.3390/cells10081923_

Round 1

Reviewer 1 Report

The latest judgments (line 68-80) of the Introduction are more appropriate for the Result section since they anticipate and give relevance to the results. The authors have to keep in mind that the introduction should  explained the purposes/objectives and hypotheses which are the basis of the research they conducted.

The methods and analysis of the obtained data are accurated and can have a relevance in the glaucoma study, but this information may not reflect the actual in vitro condition that really occurs at the level of the matrix components.

The authors said that the cells were generated as previously described  and have cited Gagen D, Filla MS, Clark R, et al. Activated avb3 integrin regulates avb5 integrin–mediated phagocytosis in trabecular meshwork cells. Invest Ophthalmol Vis Sci. 2013;54:5000-5011. But this paper referred   in turn refers to another  article of 2011 Filla MS,  et al.  Dexamethasone-associated cross-linked actin network formation in human trabecular meshwork cells involves beta3 integrin signaling. Invest Ophthalmol Vis Sci. 2011; 52: 2952–2959 for the generation of TM cells. Unfortunately also the authors of this last paper referred for the generation and characterization of TM cells two paper of Polansky et al. which are dated  1979, and 1981 (Polansky JR, et al. Human trabecular cells, I: establishment in tissue culture and growth characteristics. Invest Ophthalmol Vis Sci. 1979;18:1043–1049; Polansky JR, et al. . Studies on human trabecular cells propagated in vitro. Vis Res. 1981;21:155–160). These  bibliographical references   are too old  and have to be updated since recently several articles are focused  on the almost unique particular features of TM cells, which for their neural crest origin,  evidenced endothelial and fibroblast phenotypes. Therefore,  TM cells display   morphological features, behaviour and functions that are to be related to four different cell types: endotheliocytes, macrophages, fibroblasts and smooth muscle cells ( Stamer WD, Clark AF. The many faces of the trabecular meshwork cell. Exp Eye Res. 2017;158:112-123. doi:10.1016/j.exer.2016.07.009). So, the authors might to verify the real feature of the cells which they used in their study. To overcome this relevant issue, recently Keller et al. have provided accurate  recommendations for trabecular meshwork cell isolation, characterization and culture (Keller KE, et al. Consensus recommendations for trabecular meshwork cell isolation, characterization and culture. Experimental Eye Research. 2018;171: 164–173. doi:10.1016/j.exer.2018.03.001). After this check the authors will be sure on the real phenotype and function of the cells they have analysed. This information will gives more relevance to the   interesting results reported in this paper.   

Author Response

Reviewer #1 Comments and Suggestions for Authors

  1. Reviewer #1 felt that the English and style needed to be improved.

It is unclear what the reviewer is referring to, since the other reviewers did not to have any problems with the writing in the manuscript. Nevertheless, we have gone through the manuscript and revised some of the text that may have been difficult to follow. These areas of revised text are visible as tracked changes throughout the manuscript. Please refer to lines 206-207; 256-257; 436-437 and 643-650.

  1. The latest judgments (line 68-80) of the Introduction are more appropriate for the Result section since they anticipate and give relevance to the results. The authors have to keep in mind that the introduction should explained the purposes/objectives and hypotheses which are the basis of the research they conducted.

With all respect, it is not unusual to include a summary of the study in the last paragraph of an introduction. Many of the other articles published in Cells have a concluding introduction paragraph like ours where it summarizes the findings and states a general conclusion. However, we see the reviewer’s point and have modified the paragraph to state what the study is going to look at. Please refer to the modified final paragraph of the revised Introduction, lines 71-78.

  1. The methods and analysis of the obtained data are accurate and can have a relevance in the glaucoma study, but this information may not reflect the actual in vitro condition that really occurs at the level of the matrix components.

The purpose of this study was not to determine wat is happening at the matrix level, but to get a better understand of the transcriptional and signal transduction events that could be altering cell behavior and the matrix.

  1. The authors said that the cells were generated as previously describedand have cited Gagen D, Filla MS, Clark R, et al. Activated avb3 integrin regulates avb5 integrin–mediated phagocytosis in trabecular meshwork cells. Invest Ophthalmol Vis Sci. 2013;54:5000-5011. But this paper referred   in turn refers to another article of 2011 Filla MS, et al.  Dexamethasone-associated cross-linked actin network formation in human trabecular meshwork cells involves beta3 integrin signaling. Invest Ophthalmol Vis Sci. 2011; 52: 2952–2959 for the generation of TM cells. Unfortunately also the authors of this last paper referred for the generation and characterization of TM cells two paper of Polansky et al. which are dated  1979, and 1981 (Polansky JR, et al. Human trabecular cells, I: establishment in tissue culture and growth characteristics. Invest Ophthalmol Vis Sci. 1979; 18:1043–1049; Polansky JR, et al. Studies on human trabecular cells propagated in vitro. Vis Res. 1981;21:155–160). These bibliographical references   are too old and have to be updated since recently several articles are focused on the almost unique particular features of TM cells, which for their neural crest origin, evidenced endothelial and fibroblast phenotypes. Therefore, TM cells display   morphological features, behaviour and functions that are to be related to four different cell types: endotheliocytes, macrophages, fibroblasts and smooth muscle cells ( Stamer WD, Clark AF. The many faces of the trabecular meshwork cell. Exp Eye Res. 2017;158:112-123. doi:10.1016/j.exer.2016.07.009). So, the authors might to verify the real feature of the cells which they used in their study. To overcome this relevant issue, recently Keller et al. have provided accurate recommendations for trabecular meshwork cell isolation, characterization and culture (Keller KE, et al. Consensus recommendations for trabecular meshwork cell isolation, characterization and culture. Experimental Eye Research. 2018;171: 164–173. doi:10.1016/j.exer.2018.03.001). After this check the authors will be sure on the real phenotype and function of the cells they have analysed. This information will gives more relevance to the   interesting results reported in this paper.   

We added text in the methods to clarify the origin of the TM-1 cells and the subsequent generation of the EV, WTβ3 and CAβ3 lines. Please refer to the first paragraph of the revised Methods section lines 82-91. The immortalized parent cell line was generated from primary human trabecular meshwork established by Dr’s Jon Polansky and Jorge Alvarado who pioneered and established the criteria for TM cells in culture and in vivo. These HTM cells were steroid responsive and were used to identify myocilin (previously called TIGR). They also formed CLANs in vitro, were phagocytic, and displayed the morphological features associated with TM cells in vivo, thus meeting the criteria outlined in the Keller et al consensus paper.   As we have shown in numerous publications, the parent TM-1 cells retain these properties. They can form CLANs, are phagocytic, upregulate myocilin mRNA in response to DEX, and show a DEX-induced inhibition in phagocytosis detected in vivo. 

We would also like to point out that the reviewer’s issues that the basis for the criteria used to characterize our TM cells is outdated is unfounded. The Consensus paper that the reviewer is referring to was written in large part by Dr. Peters and it clearly cites Dr. Polanksy’s initial work (Polansky JR, et al. Human trabecular cells, I: establishment in tissue culture and growth characteristics. Invest Ophthalmol Vis Sci. 1979;18:1043–1049; Polansky JR, et al. Studies on human trabecular cells propagated in vitro. Vis Res. 1981;21:155–160) as the basis for studies that came afterward.

Reviewer 2 Report

This study describes the effects of activated αvβ3 integrin effects on the gene expression profile of human trabecular meshwork cell line in comparison with elevated levels of αvβ3 integrin. These authors have shown in several studies previously that activated αvβ3 integrin leading to decreased aqueous humor outflow through the trabecular pathway which can lead to elevated intraocular pressure, a major risk factor for glaucoma, a blinding disease. Therefore, the authors were interested in understanding how the activated αvβ3 integrin induced signaling might lead to elevated intraocular pressure. To address this, in this study, the authors have focused on the differential expression profile of transcriptome of the trabecular meshwork cells induced by the activated αVβ3 integrin.

Overall, there is a good rationale for the proposed study and the experimental design and analyses were sound and appropriate, and conclusions were based on the well validated (statistically) findings. Although, this is a rationale study, the study outcome and its impact is somewhat vague. The study finds thousands of genes exhibiting the altered expression profile under the described treatment condition. From these studies, it is difficult to draw any physiologically relevant conclusions. Yes, you always find some relevance but is it really what you wished for. Finding a needle in haystack. However, it is worth reporting the findings otherwise the question will be lingering regarding how this activated signaling protein might influence the gene expression profile and the pathways in the trabecular meshwork cells.

Although, the authors have found some interesting changes in the expression profile of TGF-beta, Wnt and cadherins, the conclusions can have some physiological significance if they can validate the observed findings with biochemical and immunohistochemical approaches.

The following analyses can improve substantially the impact of this study:

  1. Under the described experimental conditions, it would be important to determine whether there are changes in the levels of TGF-beta in the conditioned media by ELISA or other methods. If they are changes, are those changes physiologically significant?
  2. Under the described experimental conditions and cell line, does activated αvβ3 integrin induce changes in the actin cytoskeletal organization and cell adhesion? Will these findings be different when compared to the results of the cells expressing the elevated levels of αvβ3 integrin.
  3. It is not clear from the conclusions whether there is an activation or inhibition of Wnt signaling under activation of αVβ3 integrin signaling. How does inhibition of Wnt signaling leads to crosslinked actin and cell adhesion?
  4. Does activated αvβ3 integrin induce Cadherin-6 based cell-cell junctions in the trabecular meshwork cells.

The authors also should comment about what are the genes that exhibited robust changes in their expression profile (both up and down, not only the transcription factors but downstream physiological proteins and pathways) even if they are not relevant to the aqueous humor outflow. A volcano plot reveal this. Always from these global and unbiased approaches, we learn something new and unexpected. The authors can comment on these aspects rather than just emphasizing on the pathways that have been already implicated in the pathophysiology of glaucoma.   

Author Response

Reviewer #2 Comments and Suggestions for Authors

Although, the authors have found some interesting changes in the expression profile of TGF-beta, Wnt and cadherins, the conclusions can have some physiological significance if they can validate the observed findings with biochemical and immunohistochemical approaches.

The following analyses can improve substantially the impact of this study:

  1. Under the described experimental conditions, it would be important to determine whether there are changes in the levels of TGF-beta in the conditioned media by ELISA or other methods. If they are changes, are those changes physiologically significant?

We have added ELISA data showing that TGFβ2 levels are upregulated when αvβ3 integrin is activated. Please refer to the revised Methods section, lines 156-164, the revised Results section and Figure 11, lines 580-588.

Under the described experimental conditions and cell line, does activated αvβ3 integrin induce changes in the actin cytoskeletal organization and cell adhesion? Will these findings be different when compared to the results of the cells expressing the elevated levels of αvβ3 integrin.

Given the short turnaround time we were given by the journal we were unable to perform any additional experiments that would address this query. However, we have previously published that activating αvβ3 integrin causes the reorganization of the actin cytoskeleton into crosslinked actin networks (commonly called CLANs). We have also shown that activation of αvβ3 integrin impairs phagocytosis by altering GTPase signaling. Activation of this integrin does not affect cell adhesion. We previously showed, however, that activated αvβ3 integrin can be found in focal adhesions in the CAβ3 cells. These studies were cited in the manuscript.

  1. It is not clear from the conclusions whether there is an activation or inhibition of Wnt signaling under activation of αVβ3 integrin signaling. How does inhibition of Wnt signaling leads to crosslinked actin and cell adhesion?

This is an interesting question. At the moment, we do not know if Wnt signaling is activated or inhibited when avb3 integrin is activated. Figuring this out is clearly beyond the scope of this paper, but certainly something we would like to pursue in the future. Studies have shown that Wnt signaling can control TGFβ2 signaling in TM cells, however the role of the Wnt genes found in our study to be affected by activated avb3  integrin have not been studied in TM cells. Thus, it is difficult to say what affect these changes in Wnt signaling would have.

  1. Does activated αvβ3 integrin induce Cadherin-6 based cell-cell junctions in the trabecular meshwork cells.

We have not look specifically at cadherin-6 based cell-cell junctions.

The authors also should comment about what are the genes that exhibited robust changes in their expression profile (both up and down, not only the transcription factors but downstream physiological proteins and pathways) even if they are not relevant to the aqueous humor outflow. A volcano plot reveal this. Always from these global and unbiased approaches, we learn something new and unexpected. The authors can comment on these aspects rather than just emphasizing on the pathways that have been already implicated in the pathophysiology of glaucoma.   

Unfortunately, the tight turnaround time that the journal allowed us for revisions did not permit us to put together adequate volcano plots of the DGE data. We have included a supplemental file, however, that includes the DGE data that provided the basis for this entire study. The Log2 FC values for all the genes that demonstrated significant changes as determined by edgeR analysis have been ranked from lowest to highest so that readers can see what other genes were significantly changed. Please refer to the revised Methods section, lines 124-128, the Table 1 legend, lines 212-213 and line 810.

With all due respect, however, the manuscript did not entirely focus on transcription factors. We did include cytoskeleton genes, inflammatory response, cell-cell adhesion and ECM proteins that showed robust changes in their expression profile. We also discussed briefly how these changes could be involved in the pathophysiology of glaucoma. We did not expand greatly on these proteins because it would have substantially increased the size of the manuscript, which is already very large.

We would also like to point out that although we talk about TGFβ2 signaling, the emphasis of this paper is not that TGFβ2 causes glaucoma but that αvβ3 integrin signaling caused an increase in TGFβ2 mRNA and protein levels as well as changes in many proteins associated with glaucoma. Thus changes in cell-matrix interactions are likely to have a significant impact on the pathogenesis of glaucoma. This has not been previously reported and suggests that a phenomenon called “integrin switching” in which the expression of the normal integrin repertoire on the cell surface is changed or “switched” can have significant downstream effects.  Finally, our study is the first to shows that this type of switch to increased αvβ3 integrin expression and signaling could be responsible for enhanced TGFβ2 expression in POAG.

Reviewer 3 Report

The manuscript is very interesting. In my opinion, a small conclusion paragraph should be added to sum up the results.

Author Response

Reviewer #3 Comments and Suggestions for Authors

The manuscript is very interesting. In my opinion, a small conclusion paragraph should be added to sum up the results.

We have added text to both the beginning and the end of the Discussion that we believer addresses the reviewer’s concerns. Please refer to the revised Discussion, lines 718-729 and lines 798-807.

Round 2

Reviewer 1 Report

The authors revised the paper taking in account the comments and  suggestions.

Reviewer 2 Report

The authors rebuttal is satisfactory and the revision has improved the quality of manuscript .